# Sociodemographic risk factors of under-five stunting in Bangladesh: Assessing the role of interactions using a machine learning method

**Mohaimen Mansur**[1]*, **Awan Afiaz**[1], **Md. Saddam Hossain**[2]

**1** Institute of Statistical Research and Training, University of Dhaka, Dhaka, Bangladesh, **2** Department of Statistics, Research Division, Population Council, Dhaka, Bangladesh

* mmansur@isrt.ac.bd

## Abstract

This paper aims to demonstrate the importance of studying interactions among various sociodemographic risk factors of childhood stunting in Bangladesh with the help of an interpretable machine learning method. Data used for the analyses are extracted from the Bangladesh Demographic and Health Survey (BDHS) 2014 and pertain to a sample of 6,170 under-5 children. Social and economic determinants such as wealth, mother's decision making on healthcare, parental education are considered in addition to geographic divisions and common demographic characteristics of children including age, sex and birth order. A classification tree was first constructed to identify important interaction-based rules that characterize children with different profiles of risk for stunting. Then binary logistic regression models were fitted to measure the importance of these interactions along with the individual risk factors. Results revealed that, as individual factors, living in Sylhet division (OR: 1.57; CI: 1.26–1.96), being an urban resident (OR: 1.28; CI: 1.03–1.96) and having working mothers (OR: 1.21; CI: 1.02–1.44) were associated with higher likelihoods of childhood stunting, whereas belonging to the richest households (OR: 0.56; CI: 0.35–0.90), higher BMI of mothers (OR: 0.68 CI: 0.56–0.84) and mothers' involvement in decision making about children's healthcare with father (OR: 0.83, CI: 0.71–0.97) were linked to lower likelihoods of stunting. Importantly however, risk classifications defined by the interplay of multiple sociodemographic factors showed more extreme odds ratios (OR) of stunting than single factor ORs. For example, children aged 14 months or above who belong to poor wealth class, have lowly educated fathers and reside in either Dhaka, Barisal, Chittagong or Sylhet division are the most vulnerable to stunting (OR: 2.52, CI: 1.85–3.44). The findings endorse the need for tailored-intervention programs for children based on their distinct risk profiles and sociodemographic characteristics.

## Introduction

Despite many socioeconomic challenges, Bangladesh achieved remarkable successes in many areas of public health during the last two decades which include reducing child and maternal

cfm. Searching 'Bangladesh DHS, 2014' in the DHS website will provide the survey data set.

**Funding:** The author(s) received no specific funding for this work.

**Competing interests:** The authors have declared that no competing interests exist.

mortality to significantly low levels. [1]. However, high childhood malnutrition has remained as one of the lasting national health problems [2]. A key outcome of this nutrition deficiency is low height-for-age, commonly known as stunting, which reflects the aggregate repercussions of undernutrition in children since birth or even prenatal stages, indicating long-term restriction on a child's growth potential [3]. The current situation of childhood stunting in Bangladesh is alarming. According to the latest round of Bangladesh Demographic and Health Survey conducted in 2017, almost one-third of the under-five children fail to reach a standard height for their ages [4]. This figure is one of the highest in the world outside Sub-Saharan Africa, and largely overshadows the 12 percentage-point decline in the prevalence of stunting that has been achieved by Bangladesh over the last decade [4].

Bangladesh has done relatively well in reducing the prevalence of childhood stunting from 51% in 2004 to 31% in 2017. However, the resulting average annual rate of reduction (AARR) of 3.4% is still below the World Health Organization (WHO) recommended AARR of f 3.9% to meet the global target of a 40% reduction in the number of children in the world who are stunted by 2025 [5]. Only if the rapid reduction in the last few years can be maintained, Bangladesh will achieve its government's 4th Health Nutrition and Population Sector Program (HNPSP) objective of limiting the prevalence of stunting in children within 25% by the year 2022 [4]. The high prevalence together with the demand for continued high rate of reduction in childhood stunting presents a persistent threat to the national health status of children in Bangladesh. It is now well-known that in addition to fatal consequences like child mortality low height-for-age also has long-term health effects including chronic illness, disabilities, and cognitive impairment [6, 7]. Moreover, stunted children are likely to be physically, intellectually, and economically less productive in adulthood than well-nourished children [8, 9].

The long-standing problem of childhood stunting in Bangladesh has duly attracted substantial attention in academic research. A major interest of this vast literature lies in identifying the sociodemographic determinants of stunting, a key step for designing effective socioeconomic policy interventions targeted to reduce this public health problem. Risk factors that commonly appeared in many empirical studies include children's age, gender, birth order, underweight mothers, low levels of parent's education, and poor socioeconomic status, among others [2, 10–21]. Rarely addressed in these existing papers, however, is the possible complex interaction among different sociodemographic features of children. It is often the intricate interplay of multiple risk factors rather than individual factors that characterize potential risks of young-age undernourishment. For example, while nourishment is generally low in children whose families are poor or children whose mothers are not educated, it is probably even lower in children who are exposed to both these risk factors. From a policy perspective, this would mean that improving both education and economic conditions of the most vulnerable group may yield a faster and more efficient reduction in stunting.

The main argument of this paper stems from the fact that studies that ignore these possible interactions are likely to provide an inadequate understanding of the sociodemographic disparities driving childhood stunting and therefore, are likely to offer limited policy values. This study, therefore, aims to justify the merits of incorporating such interactions among potential driving factors of childhood stunting in Bangladesh by using data from a nationally representative survey. A major contribution of the paper is methodological which involves utilizing a classification tree, a method borrowed from the machine learning arena, to automatically detect the complex interactions among the common risk factors of childhood stunting. Importance of these interactions are then evaluated through the use of classical logistic regression models. Interplay of multiple factors show higher association with stunting and provide deeper insight to the persistent public health problem of the country.

## Materials and methods

### Data overview

This study used data from the Bangladesh Demographic and Health Survey (BDHS) 2014. This is a nationally representative cross-sectional survey conducted in Bangladesh in collaboration with the Demographic and Health Survey (DHS) and is operated by Measure DHS + under the authority of the National Institute of Population Research and Training (NIPORT). The survey sampling frame was based on the 2011 Bangladesh population census, and a two-stage stratified cluster sampling technique was employed in the survey. In the first stage, 600 enumeration areas (EAs), a geographic area consisting of 113 households on average, were independently selected with probability proportional size (PPS) method from all 20 sampling strata. Then using an equal probability systematic selection in the second stage, 30 households were selected from each cluster (EA). The survey resulted in 17,886 completed interviews with ever-married women between the ages of 15 to 49. Further details of the sampling procedure can be found in [22].

In addition to a number of health and nutrition relation information, the BDHS survey also collected measurements on height and weight from eligible women of reproductive ages and children under the age of 5 years. Complete and credible anthropometric and age data on 7,318 children were collected in the process [22]. After excluding the temporary (de facto) residents, children of twin births and any cases with missing values in the variables, the final sample size for this study was 6,170.

### Ethics statement

This article does not contain any studies with human participants performed by any of the authors. The Bangladesh demographic and health Surveys were approved by ICF Macro Institutional Review Board and the National Research Ethics Committee of the Bangladesh Medical Research Council. A written consent about the survey was given by participants before the interview. All identification of the respondents was dis-identified before publishing data. The secondary data sets analysed during the current study are freely available upon request from the DHS website at https://dhsprogram.com/data/available-datasets.cfm. Searching 'Bangladesh DHS, 2014' in the DHS website will provide the survey data set.

### Outcome variable

The outcome variable of this study concerns the height-for-age or stunting index which reflects the cumulative effect of constant malnourishment as well as recurrent and chronic illnesses resulting in lack of growth among children compared to their respective ages. Two standard deviations below the median (-2 SD) of the WHO reference population in terms of height-for-age indicates that the child is short for his or her age, i.e. stunted [23]. This variable was pre-calculated according to the definition provided by WHO in the BDHS 2014 child dataset.

### Independent variables

Based on the literature review and pre-analysis, the current study explored the variables that commonly appeared in many nutrition related studies on Bangladesh. At the child level, these included the age of children in months (0-59 months), sex (male, female), as well as the birth order (first born, second born, and third born or above) of the child and the decision-maker regarding child's healthcare needs (mother only, both parents, and husband and/or someone else). Independent variables related to the mother such as mother's age at the birth of her first

child (in years), number of living children (one, two, and three or more children), body mass index (BMI) level (less than 18.5, between 18.5 and 25, and greater than 25), education level (no education, primary, secondary, higher secondary or above), current employment status (currently working, not working) and mother's age difference to the father (in years) along with father's education level (no education, primary, secondary, higher secondary or above) were analyzed. It is important to mention that the 'no education' category for both parents' education implies 'no formal education.

The sociodemographic variables, related to the child's household characteristics, were also explored for association with stunting in children under 5 years of age, which included household's wealth index (poorest, poorer, middle, richer, richest), area of residence (urban, rural) and administrative division (Dhaka, Barishal, Chattogram, Khulna, Rajshahi, Rangpur, Sylhet). The wealth index quintiles were pre-calculated using principal component analysis on household assets [24] and were readily available in the dataset.

## Methodology

The analyses of the paper were carried out in two steps using a machine learning approach and a classical regression model in succession. In the first step of the analysis, a classification tree was fit to the data to identify possible interactions among predictors of childhood stunting. In the second step, the interactions identified by the trees are incorporated in a logistic regression framework in addition to individual explanatory variables, and their impact on the likelihood of stunting measured. This two-step method can be thought to be a simpler version of the *Rulefit* approach proposed in [25] where an ensemble of trees rather than a single tree is used to identify important interaction rules which are then included in a penalized regression model.

Machine learning is largely viewed as data-driven methods that can automatically detect patterns from data and use those uncovered patterns to predict future or unseen data or to make decisions under uncertainty [26]. An important member of this group, a classification tree is a popular machine learning technique. It divides the sample data into small subgroups based on simple rules involving the predictors. The rules originate from step-wise binary splits of the predictors where the most important predictor at each step (referred to as a node) is selected for splitting. The tree is produced with an aim to achieving minimum error in classification (e.g., whether a child is stunted or not) for subgroups produced at each node. A classification tree is nonparametric in nature and holds a number of advantages over parametric regression type models including minimal distributional assumptions and easier interpretation. The most important advantage, however, is the tree's inherent capability of capturing complex, such as non-linear, interactions of variables in high dimensional settings [27]. Classical regression models are not best suited to capture and analyze interactions, particularly when the important ones are not known in advance. Typically, the number of possible interactions increases drastically with the number of variables incorporated in a regression model, leading to the curse of dimensionality [28]. This eventually results in the over-parameterization of models and subsequent difficulties in estimation.

The standard classification tree techniques such as CART [29] are often criticized for biased selection of variables which have many possible splits and missing values. These problems have been rectified in a conditional inference tree framework proposed in [30] that has been adopted in the current analyses. An inherent limitation of a classification tree in general, however, is that when it is grown indefinitely, node sample sizes become smaller and the tree overfits data, meaning that it fits the noise more than the signal in the data [27]. In this analysis, the

depth of the tree was fixed to four layers to keep a balance between capturing adequate complexity and avoid too low (less than 50) sample size per node.

While trees offer flexible approaches to reveal interactions among explanatory variables from the sample data, it does not provide measurable sizes of effects of these interactions on the response (childhood stunting) or their statistical significance. To achieve these, interactions/rules were incorporated in a logistic regression setting as additional predictors. Rules yielding either a stunting prevalence of greater than 40% or less than 20% were considered suggestive of significantly high and low prevalence, respectively, and considered for entering into the regression model. These thresholds for rule selection were constructed based around the national prevalence of under-five stunting of 36% in Bangladesh [22].

Each selected rule is then converted to a binary variable taking a value of 1 when the defining criteria for that particular rule are met, and a value of 0, otherwise. Each of these binary variables is, thus, defined by unique interactions of several sociodemographic features of children. Next, two logistic regression models were estimated and compared. In the first model, only the initially selected independent variables were fitted to the stunting outcome (stunted, not stunted) and no interaction rule was considered. But in the second model, binary variables from selected rules were included along with the individual variables. Survey weights are duly applied in both the classification tree and logistic regression models for making the sample results representative of the population of interest. Findings from regression models are further adjusted for the complexity of the survey design to capture cluster- and strata-wise variations and appropriately compute the standard errors and p-values for regression estimates.

Finally, classification accuracies of the two logistic regression models are systematically evaluated for both in-sample goodness of fit and out-of-sample predictive performance. Comparisons are made by computing standard criteria such as accuracy, sensitivity, specificity and the area under the Receiver Operating Characteristic Curve (ROC). The first three criteria are calculated based on a probability threshold of 0.5. To obtain the area under the curve (AUC) for the out-of-sample ROCs, a 10-fold cross-validation approach was undertaken and the mean AUC values were reported in the findings.

To ensure the reproducibility in scientific research in cases of new discoveries, the present study followed the recommendation by [31] in identifying the significant interaction rules. That is, the proposed interaction terms will be considered significant if the corresponding p-values were less than 0.005 in the second model. All analyses were performed in R *(version 3.6.0)*. The binary logistic regression models were fitted using the *"survey"* package *(version 4.0)* and the classification tree was constructed using the *"partykit"* package *(version 1.2-7)*.

## Results

### Descriptive statistics

Table 1 presents the distribution of stunting in under-five children by their sociodemographic characteristics considered in the study. Although the age of the children was considered as a continuous numeric variable in later analysis, it was converted into a categorical variable for exploratory reasons in this table. The overall stunting prevalence was 36.5% in the study data. The results reveal that children aged two years or older were more stunted (41.8%) than younger children (28.3%). Children of higher birth order were also stunted in higher proportions (42.7%) than the first or second-born children.

The mean age of mothers at their first birth was 18.2 years with a standard deviation of 3.23 years. Fathers were mostly older than mothers with a median age difference of 8 years. Children belonging to mothers with three or more living children were more stunted (43.3%) than those households where they were the only child (30.9%) or had only one sibling (35.6%). The

**Table 1. Distribution of stunting by sociodemographic variables (categorical and continuous) among children aged 0-59 months.**

| Variables | Groups | N (%) of study participants | N (%) of stunted |
|---|---|---|---|
| **Age of child (in months)** | 0-23 months | 2432 (39.4) | 689 (28.3) |
| | 23-59 months | 3738 (60.6) | 1563 (41.8) |
| **Sex of child** | Male | 3167 (51.3) | 1175 (37.1) |
| | Female | 3003 (48.7) | 1077 (35.9) |
| **Child's birth order** | First born | 2260 (36.6) | 730 (32.3) |
| | Second born | 1885 (30.6) | 657 (34.9) |
| | Third born or above | 2025 (32.8) | 865 (42.7) |
| **Decision-maker regarding child's healthcare** | Mother only | 948 (15.4) | 319 (33.6) |
| | Both parents | 3493 (56.6) | 1252 (35.8) |
| | Husband or somebody else | 1729 (28.0) | 681 (39.4) |
| **Number of children alive to mother** | One child | 2060 (33.4) | 636 (30.9) |
| | Two children | 2122 (34.4) | 756 (35.6) |
| | Three or more | 1988 (32.2) | 860 (43.3) |
| **Mother's BMI** | Less than 18.5 | 1387 (22.5) | 633 (45.6) |
| | Between 18.5–25.0 | 3619 (58.7) | 1330 (36.8) |
| | 25 or above | 1164 (18.9) | 289 (24.8) |
| **Mother's education level** | No education | 981 (15.9) | 491 (50.1) |
| | Primary | 1727 (28.0) | 770 (44.6) |
| | Secondary | 2835 (45.9) | 866 (30.5) |
| | Higher | 627 (10.2) | 125 (19.9) |
| **Father's education level** | No education | 1564 (25.3) | 773 (49.4) |
| | Primary | 1881 (30.5) | 767 (40.8) |
| | Secondary | 1854 (30.0) | 541 (29.2) |
| | Higher | 871 (14.1) | 171 (19.6) |
| **Mother's employment status** | Not working | 4577 (74.2) | 1617 (35.3) |
| | Currently working | 1593 (25.8) | 635 (39.9) |
| **Household wealth index** | Poorest | 1359 (22.0) | 697 (51.3) |
| | Poorer | 1159 (18.8) | 476 (41.1) |
| | Middle | 1202 (19.5) | 450 (37.4) |
| | Richer | 1256 (20.4) | 394 (31.4) |
| | Richest | 1194 (19.4) | 235 (19.7) |
| **Area of residence** | Rural | 4229 (68.5) | 1623 (38.4) |
| | Urban | 1941 (31.5) | 629 (32.4) |
| **Administrative division** | Dhaka | 1081 (17.5) | 366 (33.9) |
| | Barisal | 725 (11.8) | 278 (38.3) |
| | Chattogram | 1156 (18.7) | 431 (37.3) |
| | Khulna | 687 (11.1) | 190 (27.7) |
| | Rajshahi | 756 (12.3) | 227 (30.0) |
| | Rangpur | 770 (12.5) | 272 (35.3) |
| | Sylhet | 995 (16.1) | 488 (49.0) |
| **Age of mother at first birth (mean[SD])** | | 18.2 [3.22] | 17.7 [3.02] |
| **Mother's age difference to father (median)** | | 8.0 | 7.0 |
| **Total sample size** | N | 6170 | 2252 (36.5) |

prevalence of stunting is also lower (33.6%) in cases where the mother was the sole decision-maker regarding the child's healthcare needs in comparison to cases where decisions are made by the husband or another member of the family (39.4%). The prevalence of stunting for children belonging to underweight mothers was more than 20% higher than that in children born

to mothers who were overweight (24.8%). Reduced percentages of stunting were observed for higher levels of educational attainment of both mother and father. Working mothers had a higher percentage of stunted children (39.9%). A deeper look into the data showed that the working-mother sample has almost a 9 percentage-point higher proportion of poor (poorest or poorer) individuals and a 5 percentage-point higher proportion of individuals with no formal education when compared to the non-working group. Furthermore, while the proportion of mothers with higher than secondary level education is only 1-percentage point higher in the working group, proportion of secondary-completed education is about 7-percentage point lower. The proportion of stunted children was also high at the lowest wealth quintile (51.3%) but decreased with higher levels of wealth. Percentage stunted in rural households (38.4%) was higher than the national average of 36% while that in urban households was below (32.4%). The prevalence of stunting was the highest in the Sylhet division (49%) and the lowest in the Khulna division (27.7%). Further breakdown of data revealed that the Khulna division had the lowest percentage of illiterate (not formally educated) mothers (8.3%) and one of the lowest proportions of poorest mothers (18.9%). In contrast, the proportions of uneducated and poorest mothers are among the highest (27.9% and 30.3%, respectively) in Sylhet. These facts may explain part of the observed glaring variation between stunting percentages in the two divisions.

## Classification tree

The classification tree in Fig 1 shows the interplay of important sociodemographic factors behind stunting in children under five years of age. The wealth reveals itself to be the most important predictor being placed at the top of the tree. Children belonging to middle or lower categories of wealth statuses are placed under the left branch. The age of a child is the next most important variable in this branch showing children younger than 14 months to be less

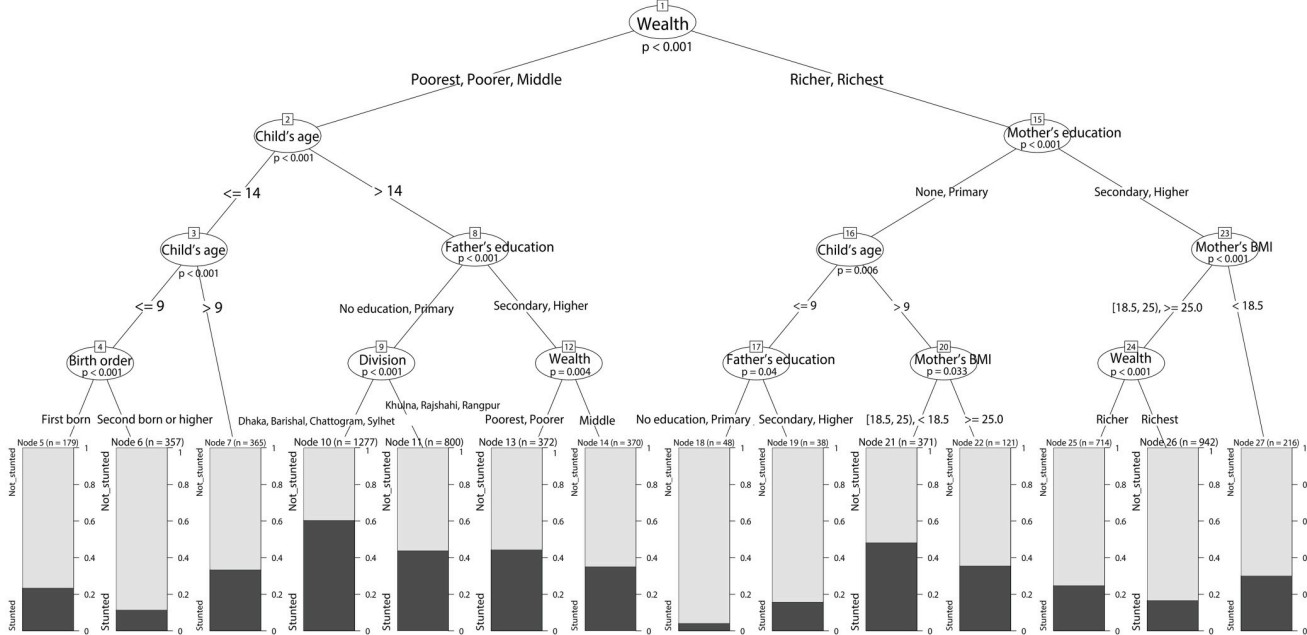

**Fig 1. Classification tree for stunting in under-five children in Bangladesh.** The value of *n* at each final node gives the node sample size and the black fill in the box below indicates proportion of stunted children under the node. The p-value at each node corresponds to a unified permutation test of hypothesis indicating independence between the response and predictors [30].

vulnerable to stunting. Other noteworthy predictors on this side of the tree were father's education, administrative division, and the birth order of the child. In this case, children older than 14 months with lowly educated fathers living in Dhaka, Barishal, Chattogram, and Sylhet divisions (node 10) seemed to be at the highest risk (around 60%) of being stunted.

The right-hand branch originated from the root of the classification tree include the wealthier households. On this side, mother's education, and then their body mass index (BMI) as well as the age of a child were notable predictors of stunting. Here, children who have mothers with no/low formal education, who are older than 9 months and have relatively lower BMI (node 21) demonstrate high risk of child stunting (around 48%). From the classification tree, we chose six interaction rules defined by six nodes (6, 10, 11, 13, 21, and 26, respectively) to be important for our analysis. The previously stated national prevalence of stunting of 36% has been utilized to determine the thresholds for these rule selections. A node was considered to be of significant importance if it showed the prevalence of stunting to be higher than 40% (high risk) or less than 20% (low risk). However, we excluded nodes 18 and 19 from further consideration on account of having considerably low sample sizes (below 50) which make reasonable statistical inference for relevant population difficult and are probably indicative of overfitting data in respective categories. The selected six nodes were then transformed into binary variables based on whether the observations satisfied all four classification criteria under each node, displayed in Table 2, and are named as "Rule- 6, 10, 11, 13, 21, and 26" respectively.

## Logistic regression models

Results from the logistic regressions are presented in Table 3. The first model which included only the sociodemographic variables in the study, exhibits that stunting was positively associated with the age of a child (OR 1.02 with 95% CI: 1.015, 1.024), corroborating the findings from the descriptive statistics. The likelihood of stunting reduced by nearly 20% when the mother was the sole decision-maker of the child's healthcare needs (OR 0.81 with 95% CI: 0.66, 0.99) or when both parents jointly made such decisions (OR 0.83 with 95% CI: 0.71, 0.97) as opposed to the father or somebody else being in charge of such decision-making. The odds of stunting reduced by 39% (p-value < .001) for mother's with a BMI greater than 25, in comparison to mothers with a BMI score between 18.5 and 25. Father's education also played a part with both secondary (OR 0.73 with 95% CI: 0.58, 0.92) and higher (OR 0.54 with 95% CI:

**Table 2. Interaction rules extracted from the classification tree and associated stunting.**

| Rule ID | Rule combinations | Node frequency | Stunting prevalence |
|---|---|---|---|
| Rule 6 | Wealth index = Poorest, Poorer, Middle & Age of child < = 14 & Age of child < = 9 & Birth order = Second born, Third born or above | 357 | 11.50% |
| Rule 10 | Wealth index = Poorest, Poorer, Middle & Age of child > 14 & Father's education level = No education, Primary & Division = Dhaka, Barisal, Chittagong, Sylhet | 1277 | 60.50% |
| Rule 11 | Wealth index = Poorest, Poorer, Middle & Age of child > 14 & Father's education level = No education, Primary & Division = Khulna, Rajshahi, Rangpur | 800 | 43.90% |
| Rule 13 | Wealth index = Poorest, Poorer, Middle & Age of child > 14 & Father's education level = Secondary, Higher & Wealth index = Poorest, Poorer | 372 | 44.40% |
| Rule 21 | Wealth index = Richer, Richest & Mother's education level = No education, Primary & Age of child > 9 & Mother's BMI = Less than 18.5, Between 18.5 to 25 | 371 | 48.20% |
| Rule 26 | Wealth index = Richer, Richest & Mother's education level = Secondary, Higher & Mother's BMI = Between 18.5 to 25, Greater than 25 & Wealth index = Richest | 942 | 16.70% |

**Table 3. Odds ratios (OR) and associated confidence intervals (CI) and p-values (p) from the two logistic regression models.**

| Vairables | Model 1: No interaction | | | Model 2: With interaction rules | | |
|---|---|---|---|---|---|---|
| | OR | 95% CI | p | OR | 95% CI | p |
| **Age of child (in months)** | 1.02 | 1.015, 1.024 | <0.001 | 1.00 | 1.00, 1.01 | 0.094 |
| **Sex of child** | | | | | | |
| Male | 1.00 | | | 1.00 | | |
| Female | 0.97 | 0.84, 1.12 | 0.666 | 0.95 | 0.83, 1.09 | 0.489 |
| **Child's birth order** | | | | | | |
| First born | 1.00 | | | 1.00 | | |
| Second born | 1.08 | 0.81, 1.45 | 0.597 | 1.06 | 0.78, 1.44 | 0.698 |
| Third born or above | 1.02 | 0.64, 1.61 | 0.941 | 1.11 | 0.70, 1.75 | 0.663 |
| **Decision-maker regarding child's healthcare** | | | | | | |
| Husband or somebody else | 1.00 | | | 1.00 | | |
| Mother only | 0.81 | 0.66, 0.99 | 0.040 | 0.82 | 0.66, 1.01 | 0.068 |
| Both parents | 0.83 | 0.71, 0.97 | 0.019 | 0.83 | 0.71, 0.97 | 0.021 |
| **Age of mother at 1st birth** | 0.98 | 0.95, 1.01 | 0.177 | 0.98 | 0.95, 1.01 | 0.236 |
| **Number of children alive to mother** | | | | | | |
| One child | 1.00 | | | 1.00 | | |
| Two children | 1.18 | 0.87, 1.58 | 0.288 | 1.34 | 0.98, 1.82 | 0.067 |
| Three or more | 1.09 | 0.68, 1.75 | 0.722 | 1.14 | 0.71, 1.81 | 0.591 |
| **Mother's BMI** | | | | | | |
| 18.5–25.0 | 1.00 | | | 1.00 | | |
| Less than 18.5 | 1.15 | 0.91, 1.45 | 0.237 | 1.11 | 0.88, 1.41 | 0.367 |
| 25 or above | 0.61 | 0.50, 0.75 | <0.001 | 0.68 | 0.56, 0.84 | <0.001 |
| **Mother's education level** | | | | | | |
| No education | 1.00 | | | 1.00 | | |
| Primary | 1.09 | 0.81, 1.46 | 0.565 | 1.05 | 0.78, 1.41 | 0.760 |
| Secondary | 0.89 | 0.66, 1.19 | 0.430 | 1.03 | 0.77, 1.38 | 0.847 |
| Higher | 0.99 | 0.64, 1.54 | 0.973 | 1.18 | 0.77, 1.83 | 0.449 |
| **Father's education level** | | | | | | |
| No education | 1.00 | | | 1.00 | | |
| Primary | 0.88 | 0.74, 1.04 | 0.124 | 0.90 | 0.75, 1.08 | 0.256 |
| Secondary | 0.73 | 0.58, 0.92 | 0.007 | 0.92 | 0.71, 1.20 | 0.545 |
| Higher | 0.54 | 0.39, 0.75 | <0.001 | 0.72 | 0.51, 1.03 | 0.070 |
| **Mother's employment status** | | | | | | |
| Not working | 1.00 | | | 1.00 | | |
| Currently working | 1.21 | 1.01, 1.44 | 0.034 | 1.21 | 1.02, 1.44 | 0.032 |
| **Mother's age difference to father** | 0.99 | 0.98, 1.00 | 0.084 | 0.99 | 0.97, 1.00 | 0.069 |
| **Household wealth index** | | | | | | |
| Poorest | 1.00 | | | 1.00 | | |
| Poorer | 0.86 | 0.71, 1.04 | 0.117 | 0.83 | 0.68, 1.01 | 0.061 |
| Middle | 0.72 | 0.59, 0.89 | 0.002 | 0.77 | 0.62, 0.95 | 0.017 |
| Richer | 0.58 | 0.45, 0.75 | <0.001 | 0.65 | 0.49, 0.87 | 0.004 |
| Richest | 0.36 | 0.26, 0.49 | <0.001 | 0.56 | 0.35, 0.90 | 0.016 |
| **Area of residence** | | | | | | |
| Rural | 1.00 | | | 1.00 | | |
| Urban | 1.29 | 1.04, 1.60 | 0.020 | 1.28 | 1.03, 1.59 | 0.028 |
| **Administrative division** | | | | | | |
| Dhaka | 1.00 | | | 1.00 | | |

*(Continued)*

**Table 3.** (Continued)

| Vairables | Model 1: No interaction | | | Model 2: With interaction rules | | |
|---|---|---|---|---|---|---|
| | OR | 95% CI | p | OR | 95% CI | p |
| Barisal | 1.11 | 0.85, 1.44 | 0.438 | 1.11 | 0.85, 1.46 | 0.433 |
| Chittagong | 1.20 | 0.97, 1.47 | 0.092 | 1.20 | 0.97, 1.48 | 0.099 |
| Khulna | 0.67 | 0.53, 0.86 | 0.002 | 0.80 | 0.61, 1.05 | 0.114 |
| Rajshahi | 0.65 | 0.52, 0.82 | <0.001 | 0.82 | 0.64, 1.05 | 0.122 |
| Rangpur | 0.88 | 0.68, 1.14 | 0.338 | 1.07 | 0.81, 1.42 | 0.632 |
| Sylhet | 1.51 | 1.23, 1.86 | <0.001 | 1.57 | 1.26, 1.96 | <0.001 |
| **Rules extracted from the classification tree** | | | | | | |
| **rule6** | | | | 0.22 | 0.15, 0.33 | <0.001 |
| **rule10** | | | | 2.52 | 1.85, 3.44 | <0.001 |
| **rule11** | | | | 1.57 | 1.19, 2.07 | 0.001 |
| **rule13** | | | | 1.64 | 1.15, 2.33 | 0.006 |
| **rule21** | | | | 2.01 | 1.41, 2.86 | <0.001 |
| **rule26** | | | | 0.65 | 0.43, 0.99 | 0.047 |

0.39, 0.75) categories showing a significantly lower likelihood of stunting compared to fathers with no education.

Interestingly, having a working mother is associated with 21% higher odds of stunting (p-value: 0.034) when compared to mothers who were not employed at the time of the survey. Children from middle, rich and, the richest households were 28%, 42%, and 64% less at risk of stunting compared to children from poorest households (p-value <.005). Furthermore, children from urban areas were significantly more likely (OR 1.29 with 95% CI: 1.04, 1.60) to be stunted than children in rural areas. Children living in Sylhet division had 51% higher risk of being stunted (p-value <.001) compared to Dhaka, whereas the burden of risk was more than 30% lower for children living in Khulna (OR 0.67 with 95% CI: 0.53, 0.86) and Rajshahi (OR 0.65 with 95% CI: 0.52, 0.82).

The second model, where the interaction rules extracted from the classification tree were included along with the sociodemographic variables, showed somewhat different results. Age of children, the mother being the sole decision-maker of child's healthcare, father's education level and administrative divisions (barring Sylhet) were no longer associated with stunting individually at 5% level of significance. However, the rest of the significant associations remained as they were with minor changes in the odds of either increasing or decreasing the likelihood of stunting.

Following the suggestion by [31], the interaction rules 6, 10, 11, and 21 were significantly associated with stunting at 0.5% level of significance. The children corresponding to rule-6, indicating those who live in the poorest, poorer or middle-wealth households, who are younger than 9 months and who rank second or higher in the birth order, were 78% less at risk of stunting than those who do not belong to this rule. According to rule-10, odds of stunting in children of lower wealth categories who are more than 14 months old, have primary or uneducated fathers and live in any one of Dhaka, Barisal, Chittagong, Sylhet divisions was 2.5 times the odds in the children who do not fall into these categories. In contrast, according to rule-11, children with same wealth, age and paternal education levels as in rule-10, exhibit an odds ratio of only 1.54 when they resided in any one of the other three divisions—Khulna, Rajshahi, or Rangpur. It is also worth noting that, children aged greater than 14 months who belonged to the poorest or poorer household but have secondary or higher educated fathers were also at

64% higher odds of stunting (rule-13 with p-value 0.006). Finally, based on rule-21, children aged greater than 9 months who belonged to the richer or richest households with mothers having little or no formal education and a BMI of 25 or less were twice as likely to be stunted as children who have different other combinations of risk factors.

## Model validation

The predictive performances of both logistic regression models were evaluated to assess whether the addition of decision tree rules as interactions in the second model played any part in improving the results. For out of sample predictions a 10-fold cross-validation procedure was followed. Table 4 shows that, when fitting in-sample data is concerned, including the decision tree rules in the model yielded a more than 18% increase in the sensitivity at the cost of a marginal drop in specificity (2.35%). Similarly, in the out-of-sample predictions, sensitivity improved by 22% with a slight drop in specificity. In both scenarios, larger areas under the ROC curve (AUC) further confirmed the benefit from adding the interaction-induced rules in the model.

## Discussion

The present analysis assessed the role of interactions between important sociodemographic factors, commonly identified in the literature, in explaining childhood stunting. The findings from the two-step analysis provided several interesting perspectives. In the exploratory part of the analysis, the classification tree identified the specific groups of under-five children with various levels of risk for stunting. Essentially, each of these different risk profiles was characterized by a unique combination of several predictively important sociodemographic features and provided noteworthy insights into the phenomenon of childhood stunting. These insights stemming from the complex interplay of multiple factors is not directly observable when looking at the pertinent factors individually. While the importance of these interactions is well perceived by the nutrition experts, supporting empirical evidences are largely undocumented.

For example, the distribution of stunted children by household wealth quintile reported in Table 1 shows that the percentage of stunted children in households belonging to the richer and richest wealth statuses were 31.4% and 19.7%, respectively. However, node 21 of the classification tree in Fig 1 (Rule 21 in Table 2) revealed that a sub-group of children from these wealth classes who were above 9 months of age and born to illiterate or lowly educated (primary education) mothers having low to moderate BMI (BMI < 25), have a much higher (48%) prevalence of stunting. Note that, the use of an interpretable machine learning method has divulged insights with more depth than what would normally be achieved.

**Table 4. Comparison of model performance.**

| Performance type | Sensitivity | Specificity | Accuracy | AUC |
|---|---|---|---|---|
| **In-sample (full sample fit)** | | | | |
| Model 1: Without interaction | 0.366 | 0.847 | 0.671 | 0.692 |
| Model 2: With interaction | 0.440 | 0.832 | 0.689 | 0.715 |
| **Out-of-sample (10-fold cross validation)** | | | | |
| Model 1: Without interaction | 0.360 | 0.850 | 0.674 | 0.694 |
| Model 2: With interaction | 0.436 | 0.824 | 0.684 | 0.712 |

Sensitivity, specificity and accuracy are computed based on a probability threshold of 0.5. Results for out-of-sample performances are averages across 10-fold cross-validations. AUC is the area under the Receiver Operating Characteristic (ROC) curve.

Similarly, vulnerable groups with analogous risk profiles could be characterized by very different combinations of risk factors with the help of classification trees. Evidently, children under node 11 of the classification tree (Rule 11 in Table 2) had only 4% lower risk of stunting than those in the aforementioned node 21, but possess noticeably dissimilar features concerning household wealth and parental education.

While classification trees are being increasingly used in many areas of public health research recently [32, 33], its applications in the field of childhood nutrition are still few [34–36]. To our knowledge, only one paper [37] used classfication trees to discover risk factors of childhood stunting in the context of Bangladesh. While the paper uses a rich set of biological, socioeconomic, and environmental information on children under two years of age, the sample data is limited only to low-income participants from a small district (smaller regions under divisions) of rural Bangladesh. Consequently, findings from the paper are not directly generalizable to the relevant child population in Bangladesh which very likely exhibits wider sociodemographic and spatial variations. We take advantage of this limitation by exploiting national-level survey data.

Findings from the classification tree of our analysis corroborate what we already know about childhood stunting in developing countries including Bangladesh while providing us with fresh perspectives. Sociodemographic variables that appeared in the tree have been largely identified as significant predictors of child growth in the literature as well as in the factor-only logistic regression model of this study (Model 1 of Table 3). The prevalence of adverse conditions and food insecurity in underprivileged households is well-documented and so is the association between poverty and stunting [20, 38]. Mother's educational attainment is an important factor for a child's nourishment [13, 39] as this corresponds to women's knowledge regarding mothering a child and their subsequent care-taking. Paternal education has been also found to be relevant [17, 40] as it is linked to increased earning and health awareness, particularly in low- and middle-income households, and complements maternal education in many cases [41].

A mother's BMI is representative of her own physical attributes, stature and nourishment. It is, therefore, not surprising that we found children having low-BMI mothers to be at higher risk of stunting. This finding is well-documented in the literature [42, 43]. Age and birth order are two other established demographic risk factors of childhood stunting. Naturally, differences in height become more prominent as children grow off their infancy. For birth order, research found that higher order children are often subject to shorter duration of breastfeeding, lower prenatal and postnatal investments and lower care-time from parents [44]. These probably explain the association of higher age and higher birth order with higher level of stunting—a result we found and also confirmed by existing research [45, 46]. Another aspect of mother that was significant in our regression analysis was her current employment status. It may seem unusual that children of working mothers appeared to be more likely to be stunted. However, as reported in the *Results* section, higher percentages of both poor and illiterate individuals in the working-mother sample may partly explain this association. However, the cross-sectional nature of the data do not allow us to venture further into the probable causes.

An interesting finding of this study is that children whose healthcare decisions are made either by the mother alone or jointly by both parents have significantly lower odds of stunting than children whose decisions are made solely by the father or someone else. This could also be a key learning in the understanding of the sociodemographic mechanism behind childhood stunting in Bangladesh and is in line with findings in the literature [47]. Individually considered, all of the discussed factors are undoubtedly important for explaining child malnutrition. However, the interplay between these factors offered a deeper understanding of the

sociodemographic effect on childhood stunting which is likely to be crucial in designing targeted intervention strategies.

In the confirmatory part of the analysis, selected groups of children with low and high risks of stunting, as identified by the tree, were incorporated into a logistic regression model (Model 2 in Table 3) together with the individual risk factors and were investigated for their possible associations with stunting. Here, all but one association were statistically significant at least at 1% level of significance. More importantly, the strength of associations of these interaction-induced risk groups with stunting, measured by odds ratio (OR), were predominantly higher (for positive associations) or lower (for negative associations) than those for individual risk factors.

As an example, the highest odds ratio for an individual variable was 1.57, observed for the Sylhet division, implying that living in this division increased the likelihood of stunting by 57% compared to the residents in the Dhaka division. In contrast, the odds ratios for positively associated rules are in the range of 1.57 to 2.52, the highest corresponding to rule 10 reported in Table 2 and implying that the odds of stunting for children living in certain divisions including Sylhet and possessing specific risk features was 2.5 times the odds for children having other characteristics. Similarly, among the individual predictors, children belonging to the richest household wealth quintile were associated with the lowest odds ratio (OR) of 0.56, while the OR for a negatively associated rule was as low as 0.22 (rule 6 in Table 2).

The combination of results from the classification tree and the logistic regression may bear valuable implications on research and policies related to stunting in under-five children. Identification of vulnerable groups and specific interactions of multiple sociodemographic factors that define these groups can aid in expanding the understanding of different aspects of stunting. Policymakers working towards the reduction of stunting may utilize this intricate knowledge in designing risk-specific tailored interventions by targeting the characterizing features.

One policy implication based on the significance of wealth-education-division interaction will pertain to identifying the poorest and lowly educated families living in high-risk divisions such as Sylhet, Barishal or Chattogram Dhaka via small-scale surveys. These are the groups most vulnerable to childhood stunting and measures should be taken to increase post-primary education and employment among them. These can be achieved by offering free or cheap higher education to these families and by creating low-capital income-generating opportunities. Based on the findings of high prevalence of stunting in relatively low-BMI and lowly educated but rich mothers, a second set of intervention programs may be formulated to actively promote maternal and child health-nutrition literacy via television and social media platforms, which are commonly accessible to the target group of potential wealthy mothers. Such targeted interventions can be quicker and more effective, but less expensive in achieving faster reduction in stunting as opposed to a nation-wide implementation of a more general policy such as making tertiary female education free of cost in the entire country. Importance of such tailored interventions are emphasized in disciplines like innovation studies [48] and nursing care [49].

The present study has a few limitations. Firstly, this study investigated only height-for-age (stunting) in children and ignored weight-for-age (overweight) and weight-for-height (wasting)—the other two widely recognized anthropometric indicators of childhood malnutrition. While a complete understanding of the drivers of childhood undernutrition will require involving all three aspects, the choice of looking at one is driven by the limited space for a single study. Secondly, the secondary data was obtained from a cross-sectional survey as opposed to a longitudinal study and so no causal relationship between the factors and the outcome could be established. Thirdly, the selection of explanatory variables is limited to those that generally appeared significant in major empirical public health studies rather than all possible inclusions from an exhaustive search of the literature. Finally, the tree depth was limited to

four layers for avoiding too low sample size per node and two nodes were excluded from analysis as a result even after exceeding the pre-specified thresholds.

## Conclusion

The aim of this paper is to assess the importance of interactions among sociodemographic risk factors of childhood stunting in Bangladesh which have been largely overlooked in the existing literature. Despite some limitations, the strength of the study lies in the novel approach of combining an interpretable machine learning method with a classical statistical model to identify these complex interactions and associated risk-groups. While wealth, area and division of residence, mother's BMI and father's education were among the important individual risk factors of stunting, certain interactions among many of these factors proved to be significantly more important. For a simple example, the combination of low wealth and low education of fathers seem to put children at higher risks of poor growth than when the two factors are considered in isolation.

When social interventions are concerned, findings of this paper emphasize importance of targeting multiple socio-economic risk factors that interact with each other in defining groups of children most vulnerable to stunting. For example, the wealth-education interaction may imply that rather than just providing safety nets (e.g., cash transfers) to poor and lowly educated pregnant mothers implementing a joint program providing them also with free education on maternal and child's health and nutrition may be more effective in reducing severe childhood stunting at later ages. At government and private policy making, these tailored policies targeting high-risk socio-economic groups may further speed-up the reduction of childhood stunting in Bangladesh. Importantly, the complementary methods demonstrated in this study can be applied to gain deeper socio-economic understanding of many other public health questions related to, for example, early childhood development, women's healthcare decision making and hygiene practices.

## Acknowledgments

The authors would like to acknowledge MEASURE DHS (Demographic and Health Surveys), National Institute for Population Research and Training (NIPORT) and USAID/Bangladesh, who allowed researchers to access the survey data for free.

## Author Contributions

**Conceptualization:** Mohaimen Mansur.

**Formal analysis:** Awan Afiaz.

**Methodology:** Mohaimen Mansur.

**Resources:** Mohaimen Mansur, Awan Afiaz.

**Supervision:** Mohaimen Mansur.

**Writing – original draft:** Mohaimen Mansur, Awan Afiaz.

**Writing – review & editing:** Mohaimen Mansur, Awan Afiaz, Md. Saddam Hossain.

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
