## [Decision Letter · Decision Letter 0]

29 Dec 2020

PONE-D-20-31337

Sociodemographic Risk Factors of Under-Five Stunting in Bangladesh: Assessing the Role of Interactions Using a Machine Learning Method

PLOS ONE

Dear Dr. Mansur,

Thank you for submitting your manuscript to PLOS ONE. After careful consideration, we feel that it has merit but does not fully meet PLOS ONE’s publication criteria as it currently stands. Therefore, we invite you to submit a revised version of the manuscript that addresses the points raised during the review process.

Reviewers feel your paper has merit for publication in PLOS One with a major revision. Address all the reviewer comments carefully and check your paper for PLOS One author guidelines before submitting the revision. 

We look forward to receiving your revised manuscript.

Kind regards,

Srinivas Goli, Ph.D.

Academic Editor

PLOS ONE

Journal Requirements:

2. To meet PLOS ONE criteria on reproducibility and data availability, please ensure that full details of the algorithms and codes designed are provided in the main text.

Additional Editor Comments:

Reviewers feels your manuscript has the merit but needs a major revision. Carefully address all the reviewer comments and write a point by point response to their comments. Please check your paper according to PLOS One authors' guidelines.

Reviewers' comments:

Reviewer's Responses to Questions

**Comments to the Author**

1. Is the manuscript technically sound, and do the data support the conclusions?

Reviewer #1: Yes

Reviewer #2: Yes

Reviewer #3: Yes

Reviewer #4: Partly

Reviewer #5: Partly

Reviewer #6: Partly

Reviewer #7: Yes

Reviewer #8: Partly

2. Has the statistical analysis been performed appropriately and rigorously? 

Reviewer #1: Yes

Reviewer #2: Yes

Reviewer #3: Yes

Reviewer #4: Yes

Reviewer #5: I Don't Know

Reviewer #6: Yes

Reviewer #7: Yes

Reviewer #8: No

3. Have the authors made all data underlying the findings in their manuscript fully available?

Reviewer #1: Yes

Reviewer #2: No

Reviewer #3: Yes

Reviewer #4: Yes

Reviewer #5: Yes

Reviewer #6: No

Reviewer #7: Yes

Reviewer #8: Yes

4. Is the manuscript presented in an intelligible fashion and written in standard English?

Reviewer #1: Yes

Reviewer #2: Yes

Reviewer #3: Yes

Reviewer #4: Yes

Reviewer #5: Yes

Reviewer #6: Yes

Reviewer #7: Yes

Reviewer #8: Yes

5. Review Comments to the Author

Reviewer #1: “Sociodemographic Risk Factors of Under-Five Stunting in Bangladesh: Assessing the Role of Interactions Using a Machine Learning Method” by Mansur M et al. is a well written manuscript with thorough analytical presentation on a very important aspect of childhood malnutrition, under 5-stunting. Childhood stunting is an important indicator of malnutrition which is prevalent in most developing countries, including Bangladesh. In spite of remarkable successes in several health indices, the rate of reduction of childhood stunting in Bangladesh is much behind expectation. The authors have tried to find out the potential driving factors of childhood stunting in Bangladesh by studying interactions among various sociodemographic risk factors with the help of an interpretable machine learning method using data from a nationally representative survey¸ Bangladesh Demographic and Health Survey 2014. Their anticipation is that by tailored-intervention programs for children based on their distinct risk profiles and sociodemographic characteristics will help to achieve the national Health Nutrition and Population Sector Program objective of limiting the prevalence of stunting in children within 25% by the year 2022 and Goal 2.2 of the SDGs to end all forms of malnutrition including stunting in children under 5 years of age by the year 2030.

The manuscript is well written, yet I have few observations as mentioned below:

The ‘Introduction’ looks unnecessarily lengthy. The first three paragraphs (Lines 2-47) are fine as the introduction for the article. In the last 4 paragraphs (Lines 49-88) the authors have explained how they identified the important interactions capable of apprehending the complex interplay between the common sociodemographic variables by utilizing the classification tree to identify groups of children with various risk profiles for stunting and then further verified it through the use of widely accepted statistical models. They have also explained its superiority over the predominant use of regression models, either in the form of linear or logistic, used in previous literatures.

The methods, authors used for the analyses, are elaborately described in ‘Methodology’ (Lines 146-206), So, such details can easily be avoided in ‘Introduction’. It would be better if the authors briefly describe their methods here before the concluding sentence of previous paragraph (Lines 45-47) and comparison of their method with the methods used in previous studies briefly in ‘Discussion’.

The prediction in lines 15-18, “Furthermore, the chronological drop in the prevalence of stunting from 51% in 2004 to 31% in 2017 translates to an average annual rate of reduction (AARR) of 3.4% which falls short of the global AARR of 3.9% required to attain World Health Organization's target to reduce by 40% the global prevalence of stunting by 2025”, might not be that straightforward. The reduction, like improvements in other health indices, probably was much faster in recent years.

It would be better to delete the sentence “Some of the recent studies are [9, 10, 11, 12, 13, 14, 15, 16, 17, 18, 19, 20, 21]” in Lines 33-34 and give the citations “9-21” after the previous sentence.

Childhood stunting (height-for-age) is an important, but not the only predictor of malnutrition; many stunting cases are either due to familial or constitutional. The anthropometric methods for measuring the nutritional status includes three widely used indicators to assess the growth of children: height-for-age, weight-for-height, and weight-for-age. Adding other parameters could strengthen the understanding of the objective of the study. It is obvious that in a single study it might not be possible to include all aspects, but at least can be mentioned as a limitation.

I am not sure why the authors have not used data from Bangladesh Demographic and Health Survey 2017.

Reviewer #2: Overall the manuscript is fine. The literature review has successfully addressed the objective of the research. The manuscript presented a nice application of machine learning method in public health research (data analysis) comparing with typical logistical model which is innovative idea. However, for a reader without having any prior knowledge of machine learning could be difficult to recognize the method.

No major issues for revision, but some minor issues to be addressed are as follows:

a) Line 225-226: “Working mothers’ had a higher percentage of stunted children (39.9%)”-Are these working mothers from lower class? Have no education? Are they working in their own area and staying with their children or they migrate to other cities leaving children at home. Need a bit explanation otherwise it looks more generalized data.

b) Line 229: The paper mentioned two specific areas (Sylhet and Khulna) of Bangladesh where the prevalence of stunning children were highest and lowest. Would you please explain which indicators were specifically worked for this variation between two areas?

c) Line 235: Figure 1, there is no figure attached.

d) In table 3, "Rules extracted from the classification tree" is a bit confusing. Is it a part logistical regression output?? If the objective of this table is to compare or find similarities then it can be placed in separate table.

e) Line 277: It would be better to understand when comparing two models, i.e. machine learning and logistic model in separate section

Other comments: Last but not the least, a simple definition with reference of Machine Learning method will be helpful for readers.

Reviewer #3: Introduction :

Line :22: Height for age?

Discussion:

Please set table 4 probably mistakenly set in discussion section to the result segment

Conclusion:

Please make it brief addressing objective

Line 419 : Can be deleted as it is not a part of objective

Reviewer #4: The article titled ‘Sociodemographic Risk Factors of Under-Five Stunting in Bangladesh: Assessing the Role of Interactions Using a Machine Learning Method’ highlighted an important public health issues in Bangladesh. I have following review comments for the authors:

a. Share few key findings in Abstract, may few numbers for the readers who does not want to read the whole article, currently it is missing.

b. 2nd sentence in Introduction, please rephrase, it sounds like stunting remained as the only public health problem in Bangladesh is high childhood malnutrition

c. Line 33-34 in Introduction section… Some of the recent studies are.. it looks like an incomplete sentence, consider including these references in previous sentence,

d. 2.2 Response variable, sound little new. Is there any reason not to use Outcome variable following the norm?

e. Conclusion in discussion section: whole conclusion section needed to be modified, currently, it sounds like an abstract of an article, sentence starting from background, methods, justification of classification tree, strengths etc..…. I would add key findings and then continue with the what is there in later part of the second paragraph.

The nodes shown in Figure 1: Although apparent criticism of the standard classification tree related to biased selection of variables supposed to be overcome by conditional inference tree framework as mentioned in the methodology section; the presented may have some inherent limitation and pose internal validity of the method: for example, mother belonged to richer and richest arm, both mother and father had no education, but their children had lowest stunting!! Stringer justification needed.

Reviewer #5: The manuscript titled “Sociodemographic Risk Factors of Under-Five Stunting in Bangladesh: Assessing the Role of Interactions Using a Machine Learning Method” is an excellent work at the current situation of Bangladesh. This analysis of Bangladesh Demographic and Health Survey (DHS) data describes a number of different factors associated with stunting using exception method of identification. However, there are some observations from my side stated below;

1. Abstract: It is surprising that none of the factors associated with stunting that were identified in the regression analysis and classification tree are mentioned in the abstract. It would be helpful to mention some of the most common risk factors in the abstract.

2. Page 3, line 4: Sentence not clear, Is there any evidence (prevalence over time) that support the statement?

3. Page 3 line 5: Stunting might be the “outcome” of nutrition deficiency rather than the “measure”.

4. Line 93: Is there any specific reason behind using the 2014 DHS dataset? Since 2014 situation has been changed as GoB and NGOs took much strategies for child and maternal nutrition improvement targeting SDG goals.

5. Line 128: Initial and exclusive breast feeding, mode of delivery, child vaccination status, complementary feeding etc. are recognized as the indicators of children's growth in many studies. Can this be included as the independent variable upon data availability? Or if there any explanation of not including those variables?

6. Line 221: By categorizing children age in to 4 to 5 groups with around 12 months interval may provide insights to observe the stunting situation among children of different age groups in regression analysis. In many studies from the similar strata found that higher prevalence of stunting prevailed among children aged 36 to 47 months (Akram R, Mostafa Kamal SM, Gaire S, Darteh EKM)

7. Table 1: Mothers education level: Does it mean "No formal education"?

8. Discussion: Can this be explained that how maternal BMI and birth order and current work status of mothers are linked to childhood stunting?

Reviewer #6: Review report of the manuscript no. PONE-D-20-31337

Abstract section, the result is not well explained because it was not fulfill to the objectives. I think it was much differ to objectives and results.

Authors could not make clear in a good way what authors were said to understand the line 4-8

28-31 lines were not obviously comprehend.

42-46 lines not satisfied for me and eventually objectives were not associated with the title.

49-55 lines explain more but it was not fruitful of the title. Even if the authors were not able to present it properly.

66-77 lines authors wrote extend but it was repeated.

Over all the introduction thrives could not well describe and inadequate.

106-107 lines what were authors said not clear for me.

Responsive variable portion wrote too much and I thought easily and curtly discuss.

Same case as independent variables elucidate too much and unnecessary.

147-206 methodology section wrote enlarge and few were repeated one or more times. Which had no resemblance to this manuscript?

What were the eligibility criteria? On the whole data sources and materials portion is pitiable presentation.

235-239 lines I didn't understand what were authors want to recognize.

241-250 lines were not clearly arrangement.

Lines 380 to 400. I didn't think these findings are relevant. The really relevant results are that the only educational level that seems to be protective is the highest level of education, as well as the only socioeconomic level is the richest.

Discussion portion could not present appropriately. In many cases authors could enlighten haphazardly. Authors could have accessible it in a simpler way. Authors end up writing it shortly.

All in all, authors could not nearby it well. But their title of the manuscript is very good and the methods that authors like are definitely appropriate for this time. But overall presentation is not appropriate of the manuscript.

Reviewer #7: It is good to see that the researchers considered a number of independent variables at children’s individual, parental and household level. These factors are useful for understanding the childhood stunting from socioeconomic point of views. However, there are some important issues such as low birth weight (LBW), infant and young child feeding (early initiation of breastfeeding, exclusive breastfeeding, breastfeeding continuation, and complementary feeding) which the nutrition professionals and policy makers are highly concerned about. Could the authors please consider including these factors into analysis or provide justification why these factors were not considered?

Abstract: Kindly mention what type of variables were considered in the statistical model. Health

Nutrition and Population Sector Program (HNPSP) mainly focuses on health and nutrition related indicators rather than socio-demographic aspects. Kindly re-phrase how the findings could be useful in national planning and policy making to reach the SDG target.

Discussion: “The complex interplay of multiple factors usually go unnoticed”—the interaction is well perceived by the nutrition experts, but the evidences are undocumented.

Conclusion: It seems the summary of every section. Kindly emphasize what the findings interpret which is well understood and sensitize relevant professionals from field level to policy level.

Reviewer #8: Response- Title needs to be enreach and shorter, type of study needs to be mentioned in the title, The findings is not elaborately mentioned in the abstract, result of OR need to be more specifically mentioned in abstract, The statistical analysis is not enough in the abstract.

6. PLOS authors have the option to publish the peer review history of their article (what does this mean?). If published, this will include your full peer review and any attached files.

Reviewer #1: No

Reviewer #2: No

Reviewer #3: No

Reviewer #4: No

Reviewer #5: No

Reviewer #6: No

Reviewer #7: **Yes: **Muhammad Abu Bakr Siddique

Reviewer #8: **Yes: **Abu Sayeed Md Abdullah

---

## [Author Response · Author response to Decision Letter 0]

10 Jul 2021

We would like to thank the reviewers for their valuable comments, which have led to an improved version of the paper. The corrections that have been made in response to the comments are provided are as below. All of the suggested corrections are marked in red, while notable deletions are marked in blue strike-through.

Reviewer #1: 

“Sociodemographic Risk Factors of Under-Five Stunting in Bangladesh: Assessing the Role of Interactions Using a Machine Learning Method” by Mansur M et al. is a well written manuscript with thorough analytical presentation on a very important aspect of childhood malnutrition, under 5-stunting. Childhood stunting is an important indicator of malnutrition which is prevalent in most developing countries, including Bangladesh. In spite of remarkable successes in several health indices, the rate of reduction of childhood stunting in Bangladesh is much behind expectation. The authors have tried to find out the potential driving factors of childhood stunting in Bangladesh by studying interactions among various sociodemographic risk factors with the help of an interpretable machine learning method using data from a nationally representative survey¸ Bangladesh Demographic and Health Survey 2014. Their anticipation is that by tailored-intervention programs for children based on their distinct risk profiles and sociodemographic characteristics will help to achieve the national Health Nutrition and Population Sector Program objective of limiting the prevalence of stunting in children within 25% by the year 2022 and Goal 2.2 of the SDGs to end all forms of malnutrition including stunting in children under 5 years of age by the year 2030. 

The manuscript is well written, yet I have few observations as mentioned below: 

The ‘Introduction’ looks unnecessarily lengthy. The first three paragraphs (Lines 2-47) are fine as the introduction for the article. In the last 4 paragraphs (Lines 49-88) the authors have explained how they identified the important interactions capable of apprehending the complex interplay between the common sociodemographic variables by utilizing the classification tree to identify groups of children with various risk profiles for stunting and then further verified it through the use of widely accepted statistical models. They have also explained its superiority over the predominant use of regression models, either in the form of linear or logistic, used in previous literatures. 

The methods, authors used for the analyses, are elaborately described in ‘Methodology’ (Lines 146-206), So, such details can easily be avoided in ‘Introduction’. It would be better if the authors briefly describe their methods here before the concluding sentence of previous paragraph (Lines 45-47) and comparison of their method with the methods used in previous studies briefly in ‘Discussion’.

Authors’ response:

We thank the reviewer for this detailed observation and apt suggestion. We have made the required changes in the ‘Introduction’ accordingly. Comparisons of results derived from the machine learning method in this study and those in literature are duly moved in the ‘Discussion’ section in the revised manuscript. All changes are marked in red.

The prediction in lines 15-18, “Furthermore, the chronological drop in the prevalence of stunting from 51% in 2004 to 31% in 2017 translates to an average annual rate of reduction (AARR) of 3.4% which falls short of the global AARR of 3.9% required to attain World Health Organization's target to reduce by 40% the global prevalence of stunting by 2025”, might not be that straightforward. The reduction, like improvements in other health indices, probably was much faster in recent years.

Authors’ response:

We thank the reviewer for this prudent observation. Indeed, while the overall AARR since 2004 is lower than required, the rate of reduction was indeed much faster in recent years. In order to clarify the problem better, we have now revised the quoted statement, and in the following statement posed the problem as ‘demand for continued rapid reduction’ rather than previously stated ‘slow rate of reduction’ [lines 23-26 in the revised manuscript]. 

It would be better to delete the sentence “Some of the recent studies are [9, 10, 11, 12, 13, 14, 15, 16, 17, 18, 19, 20, 21]” in Lines 33-34 and give the citations “9-21” after the previous sentence.

Authors’ response:

Thank you. The required changes have been made following the reviewer’s suggestion. 

Childhood stunting (height-for-age) is an important, but not the only predictor of malnutrition; many stunting cases are either due to familial or constitutional. The anthropometric methods for measuring the nutritional status includes three widely used indicators to assess the growth of children: height-for-age, weight-for-height, and weight-for-age. Adding other parameters could strengthen the understanding of the objective of the study. It is obvious that in a single study it might not be possible to include all aspects, but at least can be mentioned as a limitation.

Authors’ response:

We certainly agree with the reviewer’s comment. We have now included this point as a limitation of the study in the ‘Discussion’ section. In particular, we have added the following sentence [lines 518-523 in the revised manuscript]:

“Firstly, this study investigated only height-for-age (stunting) in children and ignored weight-for-age (overweight) and weight-for-height (wasting) - the other two widely recognized anthropometric indicators of childhood malnutrition. While a complete understanding of the drivers of childhood undernutrition will require involving all three aspects, the choice of looking at one is driven by the limited space for a single study.”

I am not sure why the authors have not used data from Bangladesh Demographic and Health Survey 2017.

We started this study almost a year ago with an aim to apply our methods to the most recent data. Unfortunately, Bangladesh Demographic and Survey (BDHS) 2017 data were not available then and were released only on December 23, 2020 – almost three months after we have submitted our manuscript. 

https://datacatalog.worldbank.org/dataset/bangladesh-demographic-and-health-survey-2017-2018

Furthermore, we would like to stress that the main objective of our study is to show the usefulness of the proposed machine learning assisted logistic regression method in capturing interactions among socio-demographic risk factors of childhood stunting, rather than provide a thorough critical analysis of current stunting situation in Bangladesh. We believe that the results of our study using BDHS 2014 data will not change much qualitatively if the recently released 2017 data were used. Moreover, the plethora of extant literature based on the 2014 BDHS dataset also plays to the advantage in the sense that the present analyses discussed in this paper is supported by sufficient literature and thereby indirectly validates the premise of the results obtained by our new methodological approach. However, we plan to investigate the possible temporal changes in a future study. 

Reviewer #2: 

Overall the manuscript is fine. The literature review has successfully addressed the objective of the research. The manuscript presented a nice application of machine learning method in public health research (data analysis) comparing with typical logistical model which is innovative idea. However, for a reader without having any prior knowledge of machine learning could be difficult to recognize the method.

No major issues for revision, but some minor issues to be addressed are as follows:

a) Line 225-226: “Working mothers’ had a higher percentage of stunted children (39.9%)”-Are these working mothers from lower class? Have no education? Are they working in their own area and staying with their children or they migrate to other cities leaving children at home. Need a bit explanation otherwise it looks more generalized data.

Authors’ response:

We thank the reviewer for the comment and enquiry. We added the following information about those working mothers following the quoted statement [lines 275-281]. 

“A deeper look into the data showed that the working-mother sample has almost 9 percentage point higher proportion of poor (poorest or poorer) individuals and 5 percentage point higher proportion of individuals with no formal education when compared to the non-working group. Furthermore, while the proportion of mothers with higher than secondary level education is only 1-percentage point higher in the working group, proportion of secondary-completed education is about 7-percentage point lower.”

The detailed cross-table results are avoided in the revised article but provided below for the reviewer’s consideration. However, we would like to clarify that other information on mothers’ workplace such as possible migration status or area of work were not available in the dataset and thus not discussed in the paper.

Working Status Wealth Index Row Total

 Poorest Poorer Middle Richer Richest 

Not working 957 806 868 960 986 4577

(Row %), (Col %) 20.9, 70.4 17.6, 69.5 19.0, 72.2 21.0, 76.4 21.5, 82.6 74.2

Working 402 353 334 296 208 1593

(Row %), (Col %) 25.2, 29.6 22.2, 30.5 21.0, 27.8 18.6, 23.6 13.1, 17.4 25.8

P-value <0.001 

Working Status Mother’s Education Row Total

 No education Primary Secondary Higher 

Not working 672 1267 2183 455 4577

(Row %), (Col %) 14.7, 68.5 27.7, 73.4 47.7,

77.0 9.9, 72.6 74.2

Working 309 460 652 172 1593

(Row %), (Col %) 19.4, 31.5 28.9, 26.6 40.9, 

23.0 10.8, 27.4 25.8

P-value <0.001 

b) Line 229: The paper mentioned two specific areas (Sylhet and Khulna) of Bangladesh where the prevalence of stunning children were highest and lowest. Would you please explain which indicators were specifically worked for this variation between two areas?

Authors’ response:

While we believe that it is difficult to identify the true indicators working behind the glaring variation between stunting percentages in the two divisions (Sylhet and Khulna) without a systematic investigation involving many potential risk factors, a quick look from our data showed that the Khulna division had the lowest percentage of illiterate (not formally educated) mothers (8.3%) and one of the lowest proportions of poorest mothers (18.9%). In contrast, the proportions of uneducated and poorest mothers are among the highest (27.9% and 30.3%, respectively) in Sylhet. These facts may explain part of the observed variation.

We added the detailed information in the text following the sentence [lines 286-291] and also provide the result below (not added in the manuscript) for the consideration of reviewer.

Division Mother’s Education Row Total

 No education Primary Secondary Higher 

Dhaka 183 313 454 131 1081

(Row %), (Col %) 16.9, 18.7 29.0, 18.1 42.0, 16.0 12.1, 20.9 17.5

Barisal 74 244 320 87 725

(Row %), (Col %) 10.2, 7.5 33.7, 14.1 44.1, 11.3 12.0, 13.9 11.8

Chattogram 161 284 618 93 1156

(Row %), (Col %) 13.9, 16.4 24.6, 16.4 53.5, 21.8 8.0, 14.8 18.7

Khulna 57 161 381 88 687

(Row %), (Col %) 8.3, 5.8 23.4, 9.3 55.5, 13.4 12.8, 14.0 11.1

Rajshahi 99 190 382 85 756

(Row %), (Col %) 13.1, 10.1 25.1, 11.0 50.5, 13.5 11.2, 13.6 12.3

Rangpur 129 200 349 92 770

(Row %), (Col %) 16.8, 13.1 26.0, 11.6 45.3, 12.3 11.9, 14.7 12.5

Sylhet 278 335 331 51 995

(Row %), (Col %) 27.9, 28.3 33.7, 19.4 33.3, 11.7 5.1, 8.1 16.1

P-value <0.001 

Division Wealth Index Row Total

 Poorest Poorer Middle Richer Richest 

Dhaka 161 153 148 261 358 1081

(Row %), (Col %) 14.9, 11.8 14.2. 13.2 13.7, 12.3 24.1, 20.8 33.1, 30.0 17.5

Barisal 170 206 157 115 77 725

(Row %), (Col %) 23.4, 12.5 28.4, 17.8 21.7, 13.1 15.9, 9.2 10.6, 6.4 11.8

Chattogram 171 172 254 292 267 1156

(Row %), (Col %) 14.8, 12.6 14.9, 14.8 22.0, 21.1 25.3, 23.2 23.1, 22.4 18.7

Khulna 130 120 157 155 125 687

(Row %), (Col %) 18.9, 9.6 17.5, 10.4 22.9, 13.1 22.6, 12.3 18.2, 10.5 11.1

Rajshahi 177 151 171 146 111 756

(Row %), (Col %) 23.4, 13.0 20.0, 13.0 22.6, 14.2 19.3, 11.6 14.7, 9.3 12.3

Rangpur 249 182 142 118 79 770

(Row %), (Col %) 32.3, 18.3 23.6, 15.7 18.4, 11.8 15.3, 9.4 10.3, 6.6 12.5

Sylhet 301 175 173 169 177 995

(Row %), (Col %) 30.3, 22.1 17.6, 15.1 17.4, 14.4 17.0, 13.5 17.4, 14.8 16.1

P-value <0.001 

c) Line 235: Figure 1, there is no figure attached.

Authors’ response:

Following PLOS ONE’s submission guidelines the figure has been attached as a separate file which should be accessible to the reviewers.

d) In table 3, "Rules extracted from the classification tree" is a bit confusing. Is it a part logistical regression output?? If the objective of this table is to compare or find similarities then it can be placed in separate table.

Authors’ response:

We apologize for the confusion that arose from the reported “Rules extracted from the classification tree” in table 3. To clarify, these are binary variables representing interactions detected in the classification tree and are part of the second logistic regression model. They did not enter the first ‘No interaction’ model, but entered the second logistic regression model as additional explanatory variables. We chose to present them in the same table in order to make a clear comparison between the two (with and without the rules/interactions) logistic regression models which we believe would be easier and beneficial to the readers.

e) Line 277: It would be better to understand when comparing two models, i.e. machine learning and logistic model in separate section

Authors’ response:

We want to clarify that the model mentioned in line 277 is actually a logistic regression model which incorporated interactions from the machine learning method (classification tree) as additional explanatory variables. This interaction-augmented logistic regression model was then compared with a plain logistic regression model without interactions. 

In this study the machine learning method is not presented as a contrasting method for logistic regression, but as a tool for effectively extracting complex interactions from data and then as a complementary method to develop more sophisticated interaction-induced logistic regression models which we posit would be an important addition to such analyses. 

Other comments: Last but not the least, a simple definition with reference of Machine Learning method will be helpful for readers.

Authors’ response:

This is an important suggestion and we have now included a simple definition of Machine Learning in the ‘Methodology’ section with reference [lines 185-188].

“Machine learning is largely viewed as data-driven methods that can automatically detect patterns from data and use those uncovered patterns to predict future or unseen data or to make decisions under uncertainty [26]. An important member of this group, a classification tree is a popular machine learning technique. It divides the sample data into small subgroups based on simple rules involving the predictors.”

Reviewer #3: 

Introduction:

Line :22: Height for age?

Authors’ response:

We apologize for the unintentional typographical mistake which has now been corrected. 

Discussion:

Please set table 4 probably mistakenly set in discussion section to the result segment

Authors’ response:

We have now moved Table 4 to the Results section following the reviewer’s advice.

Conclusion:

Please make it brief addressing objective 

Line 419: Can be deleted as it is not a part of objective

Authors’ response:

We thank the reviewer for this advice. We have now made the conclusion more concise to address specifically the objectives and deleted less relevant texts. All changes are marked in red in the revised manuscript.

Reviewer #4: 

The article titled ‘Sociodemographic Risk Factors of Under-Five Stunting in Bangladesh: Assessing the Role of Interactions Using a Machine Learning Method’ highlighted an important public health issues in Bangladesh. I have following review comments for the authors:

a. Share few key findings in Abstract, may few numbers for the readers who does not want to read the whole article, currently it is missing.

Authors’ response:

We thank the reviewer for this valuable advice and updated the abstract by adding key findings. The relevant changes are marked in red in the revised manuscript with track changes as follows:

“Results revealed that, as individual factors, living in Sylhet division (OR: 1.57; CI: 1.26 - 1.96), being an urban resident (OR: 1.28; CI: 1.03 - 1.96) and having working mothers (OR: 1.21; CI: 1.02 - 1.44) were associated with higher likelihoods of childhood stunting, whereas belonging to the richest households (OR: 0.56; CI: 0.35 - 0.90), higher BMI of mothers (OR: 0.68 CI: 0.56 - 0.84) and mothers' involvement in decision making about children's healthcare with father (OR: 0.83, CI: 0.71 - 0.97) were linked to lower likelihoods of stunting. Importantly however, risk classifications defined by the interplay of multiple sociodemographic factors showed more extreme odds ratios (OR) of stunting than single factor ORs. For example, children aged 14 months or above who belong to poor wealth class, have lowly educated fathers and reside in either Dhaka, Barisal, Chittagong or Sylhet division are the most vulnerable to stunting (OR: 2.52, CI: 1.85 - 3.44).”

b. 2nd sentence in Introduction, please rephrase, it sounds like stunting remained as the only public health problem in Bangladesh is high childhood malnutrition

Authors’ response:

We apologize for the unintentional misleading sentence and replaced it with the following:

“However, high childhood malnutrition has remained as one of the lasting national health problems.” [lines 4-5]

The change is marked in red in the revised manuscript (with track changes).

c. Line 33-34 in Introduction section… Some of the recent studies are..it looks like an incomplete sentence, consider including these references in previous sentence.

Authors’ response:

We thank the reviewer for pointing out this mistake which we have duly corrected in the revised version of the manuscript [lines 36-38].

d. 2.2 Response variable, sound little new. Is there any reason not to use Outcome variable following the norm?

Authors’ response:

We have now changed the term ‘Response variable’ to commonly used ‘Outcome variable’. The two terms are sometimes used interchangeably in statistics which is the source of the confusion.

e. Conclusion in discussion section: whole conclusion section needed to be modified, currently, it sounds like an abstract of an article, sentence starting from background, methods, justification of classification tree, strengths etc..…. I would add key findings and then continue with the what is there in later part of the second paragraph.

Authors’ response:

We thank the reviewer for this valuable advice. The conclusion is now modified accordingly in the revised manuscript and have been marked in the red in the track changes version.

The nodes shown in Figure 1: Although apparent criticism of the standard classification tree related to biased selection of variables supposed to be overcome by conditional inference tree framework as mentioned in the methodology section; the presented may have some inherent limitation and pose internal validity of the method: for example, mother belonged to richer and richest arm, both mother and father had no education, but their children had lowest stunting!! Stringer justification needed.

Authors’ response:

We appreciate the reviewer’s observation in pointing out of the inherent limitation of classification tree. While we clearly stated in ‘Classification tree’ subsection of the ‘Results’ section that we discarded two tree nodes (the above-mentioned rule is one of them) on account of having low sample sizes (less than 50), we admit that we did not clarify the justification for these exclusions which we have included now. The main argument is that nodes with lower sample sizes (resulting from overgrown trees) often overfit the data under that node and are not generalizable to relevant population. This is indeed a common criticism of trees and we now also add this limitation in the methodology section. In particular we wrote [lines 207-210]:

“An inherent limitation of a classification tree in general, however, is that when it is grown indefinitely, node sample sizes become smaller and the tree overfits data meaning that it fits the noise more than the signal in the data [27]”.

Thus, while children coming from richer and richest household and having lowly educated parents may have the lowest stunting, there are only 48 such children in the sample of over 6000 children. It is not utterly surprising to think that the risk from low education of both parents overwhelmingly outweighed the possible gain from higher wealth for this very few children, but we should not consider this as generalizable to population of such children due to small sample size. All the sample sizes for the nodes we selected for regression are over 300.

Reviewer #5: 

The manuscript titled “Sociodemographic Risk Factors of Under-Five Stunting in Bangladesh: Assessing the Role of Interactions Using a Machine Learning Method” is an excellent work at the current situation of Bangladesh. This analysis of Bangladesh Demographic and Health Survey (DHS) data describes a number of different factors associated with stunting using exception method of identification. However, there are some observations from my side stated below;

1. Abstract: It is surprising that none of the factors associated with stunting that were identified in the regression analysis and classification tree are mentioned in the abstract. It would be helpful to mention some of the most common risk factors in the abstract.

Authors’ response:

We thank the reviewer for bringing this important observation to our attention and we apologize for missing the key predictors of stunting in the abstract. We have now included them with associated odds ratios from the logistic regression analysis. The changes are marked in red in the revised manuscript.

2. Page 3, line 4: Sentence not clear, Is there any evidence (prevalence over time) that support the statement?

Authors’ response:

We have now added a reference as support for the mentioned statement in the revised manuscript [line 4].

3. Page 3 line 5: Stunting might be the “outcome” of nutrition deficiency rather than the “measure”.

Authors’ response:

We thank the reviewer for pointing out this mistake. We have now made the correction accordingly in the revised manuscript.

4. Line 93: Is there any specific reason behind using the 2014 DHS dataset? Since 2014 situation has been changed as GoB and NGOs took much strategies for child and maternal nutrition improvement targeting SDG goals.

Authors’ response:

We started this study almost a year ago with an aim to apply our methods to the most recent data. Unfortunately, Bangladesh Demographic and Survey (BDHS) 2017 data were not available then and were released only on December 23, 2020 – almost three months after we have submitted our manuscript. 

https://datacatalog.worldbank.org/dataset/bangladesh-demographic-and-health-survey-2017-2018

While prevalence of under-five stunting reduced from 36% in 2014 to 31% in 2017 (possible following drives from GoB and NGOs as mentioned by the reviewer) according the latest BDHS report, as long as socio-demographic drivers of childhood stunting are concerned, we do not expect our results/finding to change qualitatively if the newly released data were used. The identified risk factors were historically associated with under-five stunting in Bangladesh. However, we keep investigating any temporal changes between the last two survey periods as potential future research of interest. 

Also, we would like to stress that the main objective of our study is to show the usefulness of the proposed machine learning method in capturing interactions among socio-demographic risk factors of childhood stunting, rather than to provide a thorough critical analysis of the current stunting situation in Bangladesh. Moreover, the plethora of extant literature based on the 2014 BDHS dataset also plays to the advantage in the sense that the present analyses discussed in this paper is supported by sufficient literature and thereby indirectly validates the premise of the results obtained by our new methodological approach.

5. Line 128: Initial and exclusive breast feeding, mode of delivery, child vaccination status, complementary feeding etc. are recognized as the indicators of children's growth in many studies. Can this be included as the independent variable upon data?

Authors’ response:

We understand the reviewer’s rightful concern for not including the above-mentioned nutrition-related variables. Our justifications for not considering the factors are as follows:

First, we found selecting the possible set of explanatory variables for childhood stunting in our study particularly challenging. Existing studies did not help much in guiding to a common or universally accepted set of indicators, and vary substantially in terms of the number and types of variables that range from demographic and socio-economic factors to spatial and health-nutrition related factors. The main aim of our paper is to better understand the role of socio-demographic risk factors of childhood stunting through their interactions identified through a machine learning method as opposed to identify and establish a comprehensive set of best predictors of stunting. Therefore, rather than considering all possible risk factors of childhood stunting (which are numerous), in this study we focused mainly on socio-demographic determinants of under-5 stunting and other control variables which commonly appeared significantly important in many relevant empirical studies, particularly those related to Bangladesh. For this reason, we were selective and did not consider the mentioned indicators, as well as many other likely determinants of childhood stunting such as number of visits by antenatal care provider and their qualifications, hygiene practices etc. We have listed this confined (as opposed to exhaustive) selection of explanatory variables as a primary limitation of our study in the ‘Discussion’ section. 

Second, while reviewing the literature we found that many of our considered socio-demographic factors are closely related to the mentioned health and nutrition related indicators. For example, household wealth and/or mother’s education has been consistently found to be associated with complementary feeding (e.g., Dhami et al., 2019, Nkoka et al., 2018), breastfeeding practices (e.g., Rahman et al., 2020), child vaccination (e.g., Sarker et al, 2019) and choice of mode of delivery (e.g., Yaya et al., 2019) and possibly capture some of their association with stunting indirectly. 

Third, since vaccination, breastfeeding and complementary feeding requirements depend on age of children, investigating impact of these factors require age-stratified analysis which is beyond the scope of this study mainly looking at interactions of common socio-demographic factors on under-5 children as a whole.

Lastly, the interactions that would be considered in a joint sociodemographic-nutritional variables setting would be too large and require a more detailed approach. We believe such a study would require more in-depth data and analyses which would be out of the scope of this study. As mentioned previously, the main idea behind the study is to introduce a methodological contribution to the extant literature with the help of a machine learning method that is easily interpretable and allows the possibility to consider complex interactions between the variables largely ignored in typical public health studies.

We, however, understand the importance of looking at nutritional factors and plan to investigate the role of these factors and their interactions in a similar setting in a future study. We hope the reviewer would agree with our line of argument presented above. 

6. Line 221: By categorizing children age in to 4 to 5 groups with around 12 months interval may provide insights to observe the stunting situation among children of different age groups in regression analysis. In many studies from the similar strata found that higher prevalence of stunting prevailed among children aged 36 to 47 months (Akram R, Mostafa Kamal SM, Gaire S, Darteh EKM)

Authors’ response:

We thank the reviewer for this comment. We ran the regression (model with no-interaction) with children’s age as a suggested categorical variable (please see the results below) which verify the reviewer’s point: children aged 35-47 months are at the highest risk of stunting. While this is an interesting result showing possible non-linearity in age-stunting relationship we defend our treatment of age as continuous variables. 

First, the classification tree works by optimally choosing at each step the most important variable and a binary cut-off point for that variable which maximizes discrimination between the binary response values. It is also capable of capturing non-linear association between the explanatory variable and the outcome by splitting on the same variable multiple times in subsequent steps. The main reason why we did not categorize children’s age and treated it as a continuous variable in our analysis is that we wanted to allow the classification tree to automatically choose the optimal cut-off ages which will maximize discrimination between stunted and non-stunted children. Categorizing age into predefined groups would have hindered this flexibility. If there were any significantly important non-linearity in age-stunting relationship we expected it to appear in our tree, which it unfortunately did not happen. And, to keep the entire analysis consistent we treated age also as a continuous variable in the logistic regression models. Nonetheless, we understand that such subtle non-linearity may be missed if the size of the tree is kept relatively small. 

Second, the age-categorized regression did not really alter our results. The coefficients (apart from age) of the explanatory variables in the two regressions (age: categorical vs continuous) change only slightly. But importantly, the broad finding remains the same. Children of higher ages are more or less at higher risks of stunting when compared to infants (please see the classification tree in Figure 1). While the age-specific variation in stunting prevalence is interesting, we believe that it will not prove to be of high significance in addressing the aim of the paper which is exploring interaction among explanatory variables. 

Last, the use age as a continuous variable in months rather than a categorical variable with 12-month categories actually allows a more parsimonious model to be had with fewer number of parameters estimated and also permits the interpolation of results within the range of 0-59 months. 

Variables Odds Ratio 95% CI p-value

Age of child (categorized in years) 

[0,11] 1.00 

(11,23] 3.36 2.50, 4.51 <0.001

(23,35] 3.87 2.81, 5.32 <0.001

(35,47] 4.67 3.59, 6.08 <0.001

(47,59] 3.22 2.36, 4.40 <0.001

Sex of child 

Male 1.00 

Female 0.96 0.84, 1.11 0.614

Child’s birth order 

1 1.00 

2 1.00 0.74, 1.36 0.981

More than 2 0.95 0.60, 1.51 0.831

Decision maker regarding child's healthcare 

Husband or Else 1.00 

Mother 0.81 0.65, 0.99 0.045

Parents 0.83 0.71, 0.97 0.018

Age of mother at first birth (in years) 0.98 0.95, 1.01 0.282

No of children alive to mother 

1 1.00 

2 1.27 0.94, 1.72 0.125

3 or more 1.17 0.74, 1.86 0.506

Mother's BMI 

[18.5, 25) 1.00 

< 18.5 1.13 0.89, 1.44 0.325

>= 25.0 0.61 0.50, 0.75 <0.001

Mother's education level 

None 1.00 

Primary 1.05 0.78, 1.42 0.738

Secondary 0.85 0.63, 1.14 0.276

Higher 0.95 0.61, 1.48 0.821

Father's education level 

No education 1.00 

Primary 0.89 0.75, 1.05 0.168

Secondary 0.71 0.56, 0.91 0.006

Higher 0.53 0.37, 0.76 <0.001

Mother's employment status 

Not-working 1.00 

Currently working 1.21 1.02, 1.44 0.032

Mother’s age difference to the father 0.99 0.98, 1.00 0.129

Wealth index 

Poorest 1.00 

Poorer 0.89 0.73, 1.07 0.214

Middle 0.73 0.59, 0.90 0.004

Richer 0.58 0.45, 0.75 <0.001

Richest 0.35 0.25, 0.49 <0.001

Area of Residence 

Rural 1.00 

Urban 1.33 1.06, 1.65 0.012

Administrative division 

Dhaka 1.00 

Barishal 1.12 0.86, 1.45 0.414

Chattogram 1.19 0.96, 1.48 0.109

Khulna 0.67 0.52, 0.86 0.002

Rajshahi 0.66 0.52, 0.82 <0.001

Rangpur 0.88 0.67, 1.16 0.365

Sylhet 1.57 1.27, 1.94 <0.001

7. Table 1: Mothers education level: Does it mean "No formal education"?

Authors’ response:

For mother’s education in Table 1 the category “No education” indeed meant “No formal education” and has been used as the reference category in the logistic regression. We have added the following line of clarification in the ‘Independent variables’ subsection of the ‘Materials and Methods’ section where mother’s education was introduced [line 164-165]. 

“It is important to mention that the 'no education' category for both parents' education implies 'no formal education.”

8. Discussion: Can this be explained that how maternal BMI and birth order and current work status of mothers are linked to childhood stunting?

Authors’ response:

We have now added possible explanations for the association found between childhood stunting and the above-mentioned factors with references in the ‘Discussion’ section. 

The main arguments are as follows: 

1) A mother’s BMI is representative of her own physical attributes, stature and nourishment. It is, therefore, not surprising that we found children having low-BMI mothers to be at higher risk of stunting. 

2) Research found that higher order children are often subject to shorter duration of breastfeeding, lower prenatal and postnatal investments and lower care-time from parents. These probably explain the association between stunting and age and birth order - a result we found and also confirmed by existing research. 

3) It may seem unusual that children of working mothers appeared to be more likely to be stunted. However, as reported in the ‘Results’ section, higher percentages of both poor and illiterate individuals in the working-mother sample may partly explain this association. 

However, the cross-sectional nature of the data do not allow us to venture further into the probable causes. The relevant references are not listed here, but are provided in the revised manuscript.

Reviewer #6: 

Review report of the manuscript no. PONE-D-20-31337

Abstract section, the result is not well explained because it was not fulfill to the objectives. I think it was much differ to objectives and results.

Authors’ response:

We thank the reviewer for this comment. We have now included the key findings in the abstract and excluded some policy suggestions in order to make it more informative and reflective of the objectives. The main changes that are marked in red in the revised manuscript are as follows:

“Results revealed that, as individual factors, living in Sylhet division (OR: 1.57; CI: 1.26 - 1.96), being an urban resident (OR: 1.28; CI: 1.03 - 1.96) and having working mothers (OR: 1.21; CI: 1.02 - 1.44) were associated with higher likelihoods of childhood stunting, whereas belonging to the richest households (OR: 0.56; CI: 0.35 - 0.90), higher BMI of mothers (OR: 0.68 CI: 0.56 - 0.84) and mothers' involvement in decision making about children's healthcare with father (OR: 0.83, CI: 0.71 - 0.97) were linked to lower likelihoods of stunting. Importantly however, risk classifications defined by the interplay of multiple sociodemographic factors showed more extreme odds ratios (OR) of stunting than single factor ORs. For example, children aged 14 months or above who belong to poor wealth class, have lowly educated fathers and reside in either Dhaka, Barisal, Chittagong or Sylhet division are the most vulnerable to stunting (OR: 2.52, CI: 1.85 - 3.44).”

Authors could not make clear in a good way what authors were said to understand the line 4-8

Authors’ response:

We have now modified some sentences and added reference to improve the clarity of the points we made in the mentioned lines. Changes are marked in red in the revised manuscript.

28-31 lines were not obviously comprehend.

Authors’ response:

We are sorry for any unclarity from our part. In the mentioned lines we tried to explain that since childhood stunting had been a prevalent health problem in Bangladesh it attracted much academic research. Major area of investigation of these studies were identification of socio-demographic risk factors of childhood stunting.

42-46 lines not satisfied for me and eventually objectives were not associated with the title.

Authors’ response:

We have added the following subsequent texts to the mentioned lines in order to clarify our objectives in a better way. 

“A major contribution of the paper is methodological which involves utilizing a classification tree, a method borrowed from the machine learning arena, to automatically detect the complex interactions among the common risk factors of childhood stunting. Importance of these interactions are then evaluated through the use of classical logistic regression models. Interplay of multiple factors show higher association with stunting and provide deeper insight to the persistent public health problem of the country.”

Please also note that we have substantially curtailed the ‘Introduction’ by moving parts of it to the ‘Discussion section following further suggestions. We hope it now communicates our objectives in a better and precise way. 

49-55 lines explain more but it was not fruitful of the title. Even if the authors were not able to present it properly.

Authors’ response:

We thank the reviewer for this comment and apologize for the lack of clarity from our part. Please note that the entire paragraph has now been removed from the ‘Introduction’ to make it more concise. Some of the arguments in the paragraph, particularly those about relative advantages of classification tree over classical regression models, are moved to the ‘Methodology’ section. The additions are marked in red in the revised manuscript.

66-77 lines authors wrote extend but it was repeated. Over all the introduction thrives could not well describe and inadequate.

Authors’ response:

We understand the reviewer’s concern. We want to iterate that following further suggestions for improving the lack of clarity of the ‘Introduction’ and making it concise, we have modified the section substantially. Thus, the mentioned texts are removed from the ‘Introduction’ and moved to the ‘Discussion’ section of the revised manuscript.

106-107 lines what were authors said not clear for me.

Authors’ response:

We thank teh reviewer for this comment. We have modified the sentence as follows to improve its clarity:

“In addition to a number of health and nutrition relation information, the BDHS survey} also collected measurements on height and weight from eligible women of reproductive ages and children under the age of 5 years.” 

Responsive variable portion wrote too much and I thought easily and curtly discuss.

Same case as independent variables elucidate too much and unnecessary.

Authors’ response:

We understand the reviewer’s observation. We have now deleted the entire first paragraph of the ‘Response variable’ (now changed to ‘Outcome variable’ in the revised manuscript) to make the subsection more concise. However, we believe some small details were necessary to clearly introduce the nature and measurement of the various explanatory variables with their different categories/values and therefore, we could not curtail the ‘Independent variable’ section. We hope the reviewer will understand our point.

147-206 methodology section wrote enlarge and few were repeated one or more times. Which had no resemblance to this manuscript?

Authors’ response:

While we understand the reviewer’s concern, we would like to emphasize that the methodology we used in this paper, particularly the machine learning method and its application in conjunction with classical statistical models, is a relatively new concept in public health research and we wanted to make sure that it is described with adequate clarity. We, therefore, felt that the classification tree with its relative advantages and disadvantages, generation of binary rules from its interaction results, model selection criteria including cross-validation, adjustment of results for survey weights, statistical packages used etc. be touched upon and introduced in proper sequence. We hope the reviewer will understand our inability to shorten the ‘Methodology’ section despite our best efforts due to the trade-off between length and clarity. We also want to point out that we avoided description of unnecessary details of familiar methods such as the logistic regression. 

What were the eligibility criteria? On the whole data sources and materials portion is pitiable presentation.

Authors’ response:

We have reported the eligibility criteria in the ‘Data Overview’ subsection. Precisely, children under 5-years of age whose anthropometric measures of stunting (height-for-age) were available were considered for the study, but de facto residents (temporary residents), twin births and cases with missing observations on other information were excluded.

235-239 lines I didn't understand what were authors want to recognize.

Authors’ response:

In these lines we were describing left part of the classification tree originated from the root. The subsequent appearance of the variables from the root indicated the order of importance. Thus, wealth is the most important predictor of stunting as it is placed at the top. Age is the next significant predictor with children aged below 14 months have lower prevalence of stunting. In order to relate the texts to the classification tree (Figure 1) better, we now included the prevalence of stunted children (60%) for the node explained. 

241-250 lines were not clearly arrangement.

Authors’ response:

We have made the following modifications in the referred texts in order to improve clarity. 

“The right-hand branch originated from the root of the classification tree include the wealthier households. On this side, mother’s education, and then their body mass index (BMI) as well as the age of a child were notable predictors of stunting. Here, children who have mothers with no/low formal education, who are older than 9 months and have relatively lower BMI (node 21) demonstrate high risk of child stunting (around 48%).”[lines 303-307].

Changes are marked in red.

Lines 380 to 400. I didn't think these findings are relevant. The really relevant results are that the only educational level that seems to be protective is the highest level of education, as well as the only socioeconomic level is the richest.

Authors’ response:

We would like to point out that the referred texts at lines 380-392 of the original manuscript are merely policy implications of our findings rather than findings themselves, and apologize for any lack of clarity on this matter from our part. Please note that we have now modified the possible policy values in an attempt to make it more relevant to our results following suggestions from other reviewers [lines 487-502].. 

We, however, agree with the reviewer that the emphasis we gave to the established association between mother’s decision making on child’s healthcare and childhood stunting in lines 394-405 of the original manuscript may not need separate attention as such as it can be treated as a secondary result not particularly relevant to the interactions we highlighted. We have now curtailed the details of this finding and moved it to the part of the ‘Discussion’ section where individually significant variables were discussed [lines 449-454].

Discussion portion could not present appropriately. In many cases authors could enlighten haphazardly. Authors could have accessible it in a simpler way. Authors end up writing it shortly.

Authors’ response:

Please note that we have now modified the ‘Discussion’ section by including parts from the introduction and adding clarifying explanations in order to make it as coherent and relevant as possible. All the changes are marked in red in the revised manuscript.

All in all, authors could not nearby it well. But their title of the manuscript is very good and the methods that authors like are definitely appropriate for this time. But overall presentation is not appropriate of the manuscript.

Authors’ response:

We appreciate the reviewer’s encouraging words. We have now put substantial efforts to make the presentations of the paper as coherent and as organized as possible.

Reviewer #7: 

It is good to see that the researchers considered a number of independent variables at children’s individual, parental and household level. These factors are useful for understanding the childhood stunting from socioeconomic point of views. However, there are some important issues such as low birth weight (LBW), infant and young child feeding (early initiation of breastfeeding, exclusive breastfeeding, breastfeeding continuation, and complementary feeding) which the nutrition professionals and policy makers are highly concerned about. Could the authors please consider including these factors into analysis or provide justification why these factors were not considered?

Authors’ response:

We understand the reviewer’s rightful concern for not including the above-mentioned nutrition-related variables. Our justifications for not considering the factors are as follows:

First, we found selecting the possible set of explanatory variables for childhood stunting in our study particularly challenging. Existing studies did not help much in guiding to a common or universally accepted set of indicators, and vary substantially in terms of the number and types of variables that range from demographic and socio-economic factors to spatial and health-nutrition related factors. The main aim of our paper is to better understand the role of socio-demographic risk factors of childhood stunting through their interactions identified through a machine learning method as opposed to identify and establish a comprehensive set of best predictors of stunting. Therefore, rather than considering all possible risk factors of childhood stunting (which are numerous), in this study we focused mainly on socio-demographic determinants of under-5 stunting and other control variables which commonly appeared significantly important in many relevant empirical studies, particularly those related to Bangladesh. For this reason, we were selective and did not consider the mentioned indicators, as well as many other likely determinants of childhood stunting such as number of visits by antenatal care provider and their qualifications, hygiene practices etc. We have listed this confined (as opposed to exhaustive) selection of explanatory variables as a primary limitation of our study in the ‘Discussion’ section. 

Second, while reviewing the literature we found that many of our considered socio-demographic factors are closely related to the mentioned health and nutrition related indicators. For example, household wealth and/or mother’s education has been consistently found to be associated with birthweight (e.g., Khan et al., 2018, Manyeh et al., 2016), complementary feeding (e.g., Dhami et al., 2019, Nkoka et al., 2018) and breastfeeding practices (e.g., Rahman et al., 2020), and possibly capture some of their association with stunting indirectly. 

Third, since breastfeeding and complementary feeding requirements depend on age of children, investigating impact of these factors require age-stratified analysis which is beyond the scope of this study mainly looking at interactions of common socio-demographic factors on under-5 children as a whole.

Lastly, the interactions that would be considered in a joint sociodemographic-nutritional variable setting would be too large and require a more detailed approach. We believe such a study would require more in-depth data and analyses which would be out of the scope of this study. As mentioned previously, the main idea behind the study is to introduce a methodological contribution to the extant literature with the help of a machine learning method that is easily interpretable and allows the possibility to consider complex interactions between the variables largely ignored in typical public health studies.

We, however, understand the importance of looking at nutritional factors and plan to investigate the role of these factors and their interactions in a similar setting in a future study. We hope the reviewer would agree with our line of argument presented above. 

Abstract: Kindly mention what type of variables were considered in the statistical model. Health Nutrition and Population Sector Program (HNPSP) mainly focuses on health and nutrition related indicators rather than socio-demographic aspects. Kindly re-phrase how the findings could be useful in national planning and policy making to reach the SDG target.

Authors’ response:

We understand the reviewer’s point and admit that it is naïve to link the findings of the paper directly to meeting of HNPSP and SDG targets. We have now excluded this sentence from the abstract. Rather than making direct policy suggestions we emphasize targeting vulnerable children identified through distinct risk profiles and interactions of socio-demographic characteristics, learning more about their nutritional and other requirements and accordingly address multiple risk factors in designing policies to mitigate their stunting problems.

Discussion: “The complex interplay of multiple factors usually go unnoticed”—the interaction is well perceived by the nutrition experts, but the evidences are undocumented.

Authors’ response:

Here, we wanted to emphasize that the complex interplay of multiple factors is not directly observable when investigating pertinent factors individually. These interactions are not of course empirically well-investigated. However, but we apologize for the lack of clarity of the statement. We have now modified the sentence as following, appreciating acknowledgment of interactions by nutrition experts and researchers. 

“These insights stemming from the complex interplay of multiple factors is not directly observable when looking at the pertinent factors individually. While the importance of these interactions is well perceived by the nutrition experts, supporting empirical evidences are largely undocumented.”

Conclusion: It seems the summary of every section. Kindly emphasize what the findings interpret which is well understood and sensitize relevant professionals from field level to policy level.

Authors’ response:

We thank the reviewer for the suggestion. We have now curtailed and modified the conclusion to highlight the results and their policy values. All changes are highlighted in red in the revised manuscript.

Reviewer #8: 

Response- Title needs to be enreach and shorter, type of study needs to be mentioned in the title. 

Authors’ response:

We thank the reviewer for the suggestion. We, however, find it difficult to keep a balance between keeping the title short and adding more information to it, e.g., the type of study. Since the major contribution of the paper is methodological, we wanted to highlight the use of machine learning method in the title. We hope the reviewer will understand our point.

The findings is not elaborately mentioned in the abstract, result of OR need to be more specifically mentioned in abstract, The statistical analysis is not enough in the abstract.

Authors’ response:

We fully agree with the reviewer and apologize for under-reporting key results in the abstract. We have now included more detailed findings with ORs as advised which are as follows:

“Results revealed that, as individual factors, living in Sylhet division (OR: 1.57; CI: 1.26 - 1.96), being an urban resident (OR: 1.28; CI: 1.03 - 1.96) and having working mothers (OR: 1.21; CI: 1.02 - 1.44) were associated with higher likelihoods of childhood stunting, whereas belonging to the richest households (OR: 0.56; CI: 0.35 - 0.90), higher BMI of mothers (OR: 0.68 CI: 0.56 - 0.84) and mothers' involvement in decision making about children's healthcare with father (OR: 0.83, CI: 0.71 - 0.97) were linked to lower likelihoods of stunting. Importantly however, risk classifications defined by the interplay of multiple sociodemographic factors showed more extreme odds ratios (OR) of stunting than single factor ORs. For example, children aged 14 months or above who belong to poor wealth class, have lowly educated fathers and reside in either Dhaka, Barisal, Chittagong or Sylhet division are the most vulnerable to stunting (OR: 2.52, CI: 1.85 - 3.44).”

”

References

Dhami, M.V., Ogbo, F.A., Osuagwu, U.L. and Agho, K.E., 2019. Prevalence and factors associated with complementary feeding practices among children aged 6–23 months in India: a regional analysis. BMC public health, 19(1), pp.1-16.

Khan, J.R., Islam, M.M., Awan, N. and Muurlink, O., 2018. Analysis of low birth weight and its co-variants in Bangladesh based on a sub-sample from nationally representative survey. BMC pediatrics, 18(1), pp.1-9.

Manyeh, A.K., Kukula, V., Odonkor, G., Ekey, R.A., Adjei, A., Narh-Bana, S., Akpakli, D.E. and Gyapong, M., 2016. Socioeconomic and demographic determinants of birth weight in southern rural Ghana: evidence from Dodowa Health and Demographic Surveillance System. BMC pregnancy and childbirth, 16(1), pp.1-9.

Nkoka, O., Mhone, T.G. and Ntenda, P.A., 2018. Factors associated with complementary feeding practices among children aged 6–23 mo in Malawi: an analysis of the Demographic and Health Survey 2015–2016. International health, 10(6), pp.466-479.

Rahman, M.A., Khan, M.N., Akter, S., Rahman, A., Alam, M.M., Khan, M.A. and Rahman, M.M., 2020. Determinants of exclusive breastfeeding practice in Bangladesh: Evidence from nationally representative survey data. Plos one, 15(7), p.e0236080.

Sarker, A.R., Akram, R., Ali, N. and Sultana, M., 2019. Coverage and factors associated with full immunisation among children aged 12–59 months in Bangladesh: insights from the nationwide cross-sectional demographic and health survey. BMJ open, 9(7), p.e028020.

Yaya, S., Bishwajit, G. and Gunawardena, N., 2019. Socioeconomic factors associated with choice of delivery place among mothers: a population-based cross-sectional study in Guinea-Bissau. BMJ global health, 4(2), p.e001341.

---

## [Decision Letter · Decision Letter 1]

16 Aug 2021

Sociodemographic Risk Factors of Under-Five Stunting in Bangladesh: Assessing the Role of Interactions Using a Machine Learning Method

PONE-D-20-31337R1

Dear Dr. Mansur,

We’re pleased to inform you that your manuscript has been judged scientifically suitable for publication and will be formally accepted for publication once it meets all outstanding technical requirements.

Kind regards,

Srinivas Goli, Ph.D.

Academic Editor

PLOS ONE

Additional Editor Comments (optional):

Considering my own reading of the paper and reviewers opinion, I am in favour of recommending this paper.

Reviewers' comments:

Reviewer's Responses to Questions

**Comments to the Author**

1. If the authors have adequately addressed your comments raised in a previous round of review and you feel that this manuscript is now acceptable for publication, you may indicate that here to bypass the “Comments to the Author” section, enter your conflict of interest statement in the “Confidential to Editor” section, and submit your "Accept" recommendation.

Reviewer #1: All comments have been addressed

Reviewer #2: All comments have been addressed

Reviewer #3: All comments have been addressed

2. Is the manuscript technically sound, and do the data support the conclusions?

Reviewer #1: Yes

Reviewer #2: Yes

Reviewer #3: Yes

3. Has the statistical analysis been performed appropriately and rigorously? 

Reviewer #1: Yes

Reviewer #2: Yes

Reviewer #3: Yes

4. Have the authors made all data underlying the findings in their manuscript fully available?

Reviewer #1: Yes

Reviewer #2: No

Reviewer #3: Yes

5. Is the manuscript presented in an intelligible fashion and written in standard English?

Reviewer #1: Yes

Reviewer #2: Yes

Reviewer #3: (No Response)

6. Review Comments to the Author

Reviewer #1: The authors have adequately addressed my comments raised in the previous round of review. Only one observation that needs to be addressed mentioned below.

"Weight-for-age does not only indicate overweight, but also underweight." (Line 519 of revised manuscript

Reviewer #2: The manuscript has been improved a lot than last time I reviewed. In general, it is interesting and good to understand child stunting in Bangladesh.

The authors have clearly addressed all the comments made on the manuscript.

Reviewer #3: Author addressed reviewer's comments up to his full satisfaction. This manuscript can be accepted for publication.

7. PLOS authors have the option to publish the peer review history of their article (what does this mean?). If published, this will include your full peer review and any attached files.

Reviewer #1: **Yes: **A S M Nawshad Uddin Ahmed

Reviewer #2: No

Reviewer #3: No

---

## [Editor Report · Acceptance letter]

23 Aug 2021

PONE-D-20-31337R1 

Sociodemographic Risk Factors of Under-Five Stunting in Bangladesh: Assessing the Role of Interactions Using a Machine Learning Method 

Dear Dr. Mansur:

I'm pleased to inform you that your manuscript has been deemed suitable for publication in PLOS ONE. Congratulations! Your manuscript is now with our production department. 

Kind regards, 

on behalf of

Dr. Srinivas Goli 

Academic Editor

PLOS ONE